# A metal-trap tests and refines blueprints to engineer cellular protein metalation with different elements

Sophie E. Clough [1,2,3], Tessa R. Young [1,2,3], Emma Tarrant[1,2], Andrew J. P. Scott [1,2], Peter T. Chivers [1,2], Arthur Glasfeld [1,2] & Nigel J. Robinson [1,2] ✉

It has been challenging to test how proteins acquire specific metals in cells. The speciation of metalation is thought to depend on the preferences of proteins for different metals competing at intracellular metal-availabilities. This implies mis-metalation may occur if proteins become mis-matched to metal-availabilities in heterologous cells. Here we use a cyanobacterial Mn$^{II}$-cupin (MncA) as a metal trap, to test predictions of metalation. By re-folding MncA in buffered competing metals, metal-preferences are determined. Relating metal-preferences to metal-availabilities estimated using cellular metal sensors, predicts mis-metalation of MncA with Fe$^{II}$ in *E. coli*. After expression in *E. coli*, predominantly Fe$^{II}$-bound MncA is isolated experimentally. It is predicted that in metal-supplemented viable cells metal-MncA speciation should switch. Mn$^{II}$-, Co$^{II}$-, or Ni$^{II}$-MncA are recovered from the respective metal-supplemented cells. Differences between observed and predicted metal-MncA speciation are used to refine estimated metal availabilities. Values are provided as blueprints to guide engineering biological protein metalation.

A purpose of this research is to test explanations of how proteins acquire different metals in cells: the speciation of metalation in biology (Supplementary Fig. 1). Metalloenzyme catalysis is mostly metal-specific. Yet metalloproteins typically bind one or more wrong metals in preference to the cognate metal(s)[1–3]. Nascent metal sites in proteins are flexible such that non-cognate metals can bind non-conservatively by using a subset of the native ligands, by recruiting additional ligands and/or by adopting non-cognate coordination geometries[4]. With such limited constraint the order of metal binding commonly follows the Irving-Williams series (Fig. 1a)[2,5]. An exception is where there has been prior structural organisation for example via cooperativity in di-metal sites[6,7]. Here we quantify the binding preferences of a protein (MncA) which kinetically traps metals during folding[8]. We then use MncA to establish if protein metalation can be correctly predicted, then predictably adjusted, and finally to refine estimates of intracellular metal availability.

In combination with the metal-binding preferences of proteins, a second factor determining metal-protein speciation is metal availability[8]. Metal partitioning can occur directly from pools of labile, exchangeable and hence available metals to the protein of interest, or to an assembly pathway for a small molecule cofactor (such as Fe$^{II}$ into heme or Co$^{II}$ into vitamin B$_{12}$), or to a metallochaperone that delivers to an assembly pathway or to the protein of interest[9–18]. Here we explore the initial partitioning step from labile, available intracellular metal pools.

The significance of intracellular metal availability to metal-protein speciation emerged over several decades. When expressed in a cyanobacterial cell a Ni$^{II}$- and Co$^{II}$-responsive DNA-binding metal-sensor from *Mycobacterium tuberculosis* solely responded to Co$^{II}$[19]. This change in specificity was attributed to Ni$^{II}$ being insufficiently available inside viable cyanobacteria to metalate NmtR. Similarly, Fe$^{II}$-responsive DtxR from *Corynebacterium diphtheriae* gained responsiveness to Mn$^{II}$

[1]Department of Biosciences, University of Durham, Durham, UK. [2]Department of Chemistry, University of Durham, Durham, UK. [3]These authors contributed equally: Sophie E. Clough, Tessa R. Young. ✉e-mail: nigel.robinson@durham.ac.uk

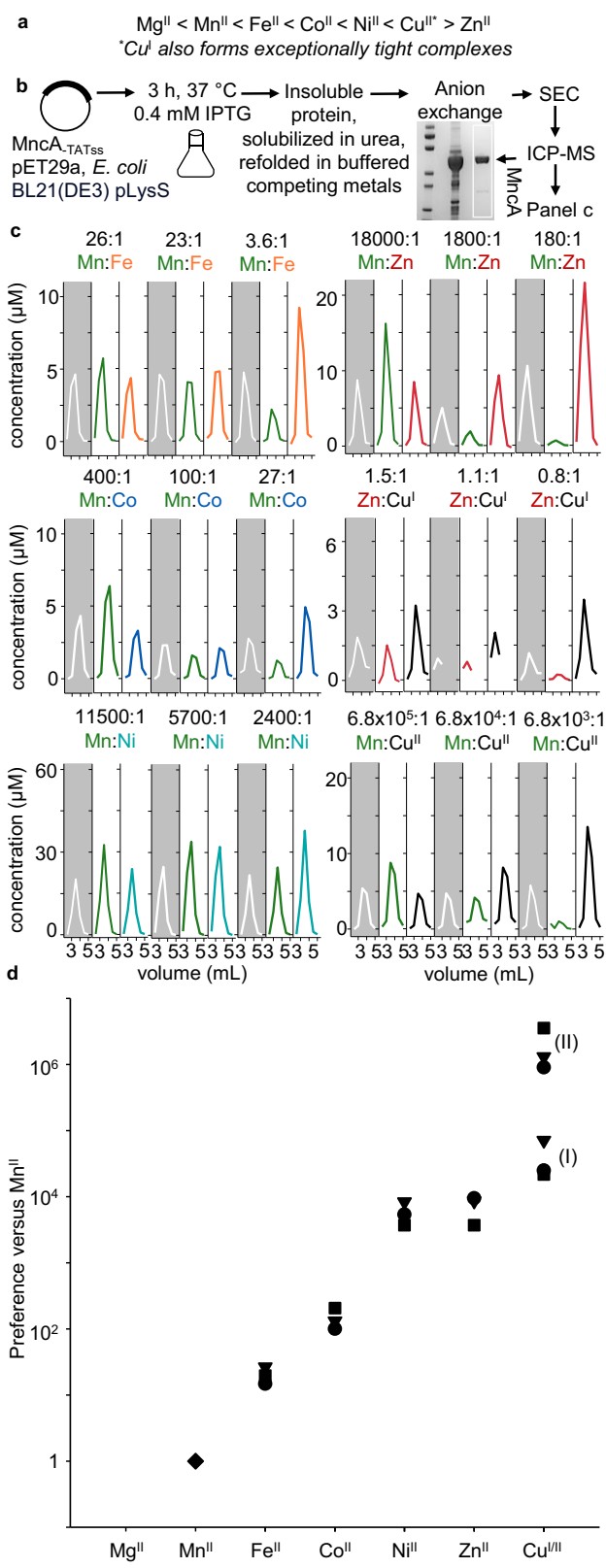

**a**

$$Mg^{II} < Mn^{II} < Fe^{II} < Co^{II} < Ni^{II} < Cu^{II*} > Zn^{II}$$
*$^*Cu^I$ also forms exceptionally tight complexes*

**b** MncA$_{-TATss}$ pET29a, *E. coli* BL21(DE3) pLysS → 3 h, 37 °C 0.4 mM IPTG → Insoluble protein, solubilized in urea, refolded in buffered competing metals → Anion exchange → SEC → ICP-MS → Panel c

**Fig. 1 | Relative metal preferences of MncA folded in competing buffered metals. a** The Irving-Williams order of formation of complexes with metals, as the ionic states of exchangeable metals in the cytosol. From weak (left) to tight, noting reversal of arrow after copper. **b** Protocol for recovery of unfolded MncA from inclusion bodies after high-level expression in *E. coli* BL21(DE3) pLysS for 3 h using pET29a$_{MncA}$ at 37 °C, followed by folding denatured MncA via dropwise dilution into a large volume of urea-free buffer containing pairs of buffered and competing metal ions. Folded MncA was concentrated by anion exchange and eluted with single-step high-salt buffer. Metal-bound MncA was separated from unbound metal by SEC (PD-10 column), and metal analysed by ICP-MS. Gel inset shows, from left, size markers (from top, 100, 70, 55, 35, 25, 15 kDa), overloaded urea-solubilised MncA and MncA eluted from Q-Sepharose (full gel Supplementary Fig. 2a). **c** Metal contents of MncA-containing fractions after folding in excesses of competing metals buffered to different ratios of availabilities (as in panel **b**, buffers in Supplementary Table 1 prepared with assistance from Supplementary Data 1). Ratios of competing metals above each panel. The concentration of MncA approximated from $A_{280nm}$ ($\varepsilon = 120,000\ M^{-1}$). Stoichiometries approximate 2:1 but proportional metalation was calculated from total metal as in Table 1. Experiments with Cu$^I$ and Fe$^{II}$ performed in an anaerobic chamber, metal stocks prepared freshly, hydroxylamine (1 mM) used to sustain reduced copper. Fe$^{II}$ and Cu$^I$ confirmed to be > 95% reduced via reaction with excess ferrozine or BCA respectively. *n* = 3 independent experimental replicates shown. **d** Quantified preferences of MncA for different divalent metals, plus monovalent copper, relative to Mn$^{II}$ (*n* = 3 independent experiments, triangles, squares, and circles depict replicates in (**c**), (I) and (II) denote forms of copper), determined from the elution profiles in (**c**). The order of preferences follows the Irving-Williams series as in (**a**). MncA prefers all non-cognate metals over Mn$^{II}$. Source data are provided as a Source Data file.

Zn$^{II}$ versus Mn$^{II}$, the non-cognate metals, copper or Zn$^{II}$, bound to MncA[8]. While the copper-cupin is secreted unfolded via the Sec-system, the Mn$^{II}$-cupin is a Tat-substrate which folds and kinetically traps the less competitive metal in the cytosol before secretion. Thus, metal availability at the site of protein folding determines metal speciation, and Mn$^{II}$ must be more available than Cu$^I$ or Zn$^{II}$ in the cyanobacterial cytosol[8]. MncA is an Mn$^{II}$-dependent oxalate decarboxylase (inactive with copper or Zn$^{II}$), and the related cytosolic OxdC from *Bacillus subtilis* trapped either Mn$^{II}$ or Co$^{II}$ in *E. coli* depending on media metal supplementation[21]. MncA thus offers enticing opportunities to directly interrogate mechanisms and predictions of metalation because (1) kinetically trapped metals are unlikely to exchange during purification, and so MncA should faithfully report its in vivo metalation state, and (2) precedent suggests it could be possible to switch the speciation of MncA metalation in *E. coli* and discover if various metalation states are predictable.

We recently developed a metalation calculator which accounts for inter-metal competition within cells[22]. The calculations use estimations of intracellular metal availabilities derived from thermodynamically calibrated responses of the cells' DNA-binding metal-sensing transcriptional regulators (Supplementary Fig. 1)[23,24]. Availabilities are described as free energies consistent with bound but labile metals capable of rapid associative ligand exchange with the protein of interest[23,25,26]. By using estimates of intracellular availabilities standardised to the mid-points of the sensor ranges (representing idealised cells) the cognate metals of four exemplar proteins have been correctly decoded[22,23]. These data encourage a view that our understanding of the speciation of metalation is (broadly) correct. However, unlike in idealised cells, metal sensors will be at different positions in their ranges in actual cells depending on growth conditions. The abundance of transcripts encoded by metal-sensor regulated genes was estimated by qPCR and then calibrated to estimate metal availabilities inside *E. coli* grown in un-supplemented medium[27]. To date, calculations of protein metal speciation have only been experimentally tested indirectly in *E. coli* engineered to manufacture vitamin B$_{12}$[22,27,28]. Co$^{II}$-dependent production of B$_{12}$ was thus measured as a proxy for the metalation states of the Co$^{II}$ metallochaperone CobW, and the Co$^{II}$ chelatase CobNST. This work now

when expressed in *Bacillus subtilis*, again attributed to different intracellular Mn$^{II}$ availabilities in the different cells[20]. Two metal-binding cupins, Mn$^{II}$-MncA and Cu$^{II}$-CucA, were discovered in the periplasm of a cyanobacterium (*Synechocystis* PCC 6803), notably binding metals from different ends of the Irving-Williams series yet exploiting similar folds and identical metal-binding residues[8]. When folded in vitro in similar amounts of competing copper versus Mn$^{II}$, or

tests predictions directly by using cyanobacterial MncA to read out in vivo metalation when expressed in *E. coli* cells under different growth conditions.

Here we determine whether metalation calculations correctly predict mis-metalation speculated to occur in engineered cells where metal availabilities are mismatched to the metal preferences of heterologously expressed proteins. In cells supplemented with cobalt, nickel and manganese, residual differences between predicted and observed MncA metalation are also used to refine estimated availabilities of other metals, and we test if changes in metal atoms cell$^{-1}$ align with these refinements. We explore the mechanism by which exposure to one metal (eg cobalt) can change availability of another (eg Fe$^{II}$). Calculators are included to enable use of MncA to probe metal availabilities under other growth conditions and in other cell types. Efforts are being made to engineer proteins that overcome the Irving-Williams series[29]. These approaches place constraints on, for example, bi-metallic catalytic centres. In contrast, here we show how biology can be exploited in a predictable way to overcome the challenge presented by the Irving-Williams series, thus expanding the repertoire of metal-driven biocatalysis that can be predictably utilised. Calculators are provided to guide the optimisation of protein metalation with different metals.

## Results

### Metal preferences at folding and trapping can also follow the Irving-Williams series

The first objective was to measure the metal-binding preferences of MncA in vitro. It is not feasible to measure affinities because Mn$^{II}$ is entrapped within the folded protein such that off-rates become negligible[8]. Instead, relative preferences during folding were determined. Rapid, high-level, expression of MncA (minus secretion signal peptide) in *E. coli* produces MncA-containing inclusion bodies from which unfolded apo-protein can be recovered[8]. MncA was thus expressed in *E. coli* BL21(DE3) pLysS, and urea-solubilised MncA, shown in Fig. 1b, was refolded by dilution into urea-free buffer. Refolding solutions contained pairs of competing metals buffered with NTA (or histidine for Ni$^{II}$ competitions) as in Supplementary Table 1, formulated via Supplementary Note 1 using the provided calculator (Supplementary Data 1). Competitions involving Fe$^{II}$ and Cu$^{I}$ were performed in an anaerobic chamber with N$_2$-purged buffers and metal stocks confirmed > 95% reduced immediately prior to use. Metals were unbuffered in competitions between Cu$^{I}$ and Zn$^{II}$. Refolded MncA (Fig. 1b), recovered by anion exchange chromatography, was resolved from unbound metal by size exclusion chromatography (SEC) with fractions (0.5 mL) analysed for MncA by UV absorbance and metals by ICP-MS.

The proportion of each metal acquired by MncA was determined from the chromatograms in Fig. 1c. Challenges in generating NTA-buffered Ni$^{II}$ competitions initially led to Ni$^{II}$ competitions being performed without buffer before employing histidine-buffers. An extra replicate of histidine-buffered Ni$^{II}$-competition was also performed (Supplementary Fig. 2, Supplementary Table 2). Competition between Mn$^{II}$ and bicinchoninic acid (BCA) buffered Cu$^{I}$ confirmed that MncA has $< 4 \times 10^7$-fold preference for Cu$^{I}$, consistent with the determined $4 \times 10^4$-fold preference (Supplementary Fig. 2d, Table 1). Preferences of metal-binding to MncA at folding relative to Mn$^{II}$ were calculated from Table 1 to generate Fig. 1d. The order of binding follows the Irving-Williams series (Fig. 1a). The exchangeable forms of metals in the cytosol are thought to be divalent except copper, which is monovalent. Fig. 1d illustrates the challenge to predict the metalation states of proteins in vivo and to decode cognate metals, since here Mn$^{II}$ seems least likely.

**Table 1 | Preferences for metals trapped by MncA relative to Mn$^{II}$**

| Metal1 | Metal2 | [Metal1]$_{free}$/ [Metal2]$_{free}$$^a$ | Metal1 (%)$^b$ | Metal2 (%)$^b$ | Metal1/Metal2 | Preference$^c$ | Average preference | $\Delta G_{MP}$$^e$ ($^{kJ}/_{mol}$) |
|---|---|---|---|---|---|---|---|---|
| Mn$^{II}$ | Fe$^{II}$ | $2.59 \times 10^1$ | 56.7 | 43.3 | 1.31 | $1.98 \times 10^1$ | | |
| Mn$^{II}$ | Fe$^{II}$ | $2.26 \times 10^1$ | 45.9 | 54.1 | 0.85 | $2.67 \times 10^1$ | $2.1(\pm 0.6) \times 10^1$ | −50.8 |
| Mn$^{II}$ | Fe$^{II}$ | $3.58 \times 10^0$ | 19.3 | 80.7 | 0.24 | $1.50 \times 10^1$ | | |
| Mn$^{II}$ | Co$^{II}$ | $2.52 \times 10^1$ | 20.1 | 79.9 | 0.25 | $1.00 \times 10^2$ | | |
| Mn$^{II}$ | Co$^{II}$ | $1.00 \times 10^2$ | 43.1 | 56.9 | 0.76 | $1.32 \times 10^2$ | $1.5(\pm 0.5) \times 10^2$ | −55.6 |
| Mn$^{II}$ | Co$^{II}$ | $4.00 \times 10^2$ | 65.9 | 34.1 | 1.93 | $2.07 \times 10^2$ | | |
| Mn$^{II}$ | Ni$^{II}$ | $1.15 \times 10^4$ | 57.8 | 42.2 | 1.37 | $8.43 \times 10^3$ | | |
| Mn$^{IIf}$ | Ni$^{II}$ | $3.73 \times 10^4$ | 76.8 | 23.2 | 3.31 | $1.13 \times 10^4$ | $7.2(\pm 3.3) \times 10^3$ | −65.3 |
| Mn$^{II}$ | Ni$^{II}$ | $2.41 \times 10^3$ | 39.3 | 60.7 | 0.65 | $3.73 \times 10^3$ | | |
| Mn$^{II}$ | Ni$^{II}$ | $5.73 \times 10^3$ | 51.3 | 48.7 | 1.06 | $5.43 \times 10^3$ | | |
| Mn$^{II}$ | Zn$^{II}$ | $1.83 \times 10^4$ | 65.9 | 34.1 | 1.93 | $9.46 \times 10^3$ | | |
| Mn$^{II}$ | Zn$^{II}$ | $1.84 \times 10^3$ | 17.3 | 82.7 | 0.21 | $8.76 \times 10^3$ | $8.0(\pm 2.0) \times 10^3$ | −65.5 |
| Mn$^{II}$ | Zn$^{II}$ | $1.83 \times 10^2$ | 3.1 | 96.9 | 0.03 | $5.71 \times 10^3$ | | |
| Mn$^{II}$ | Cu$^{II}$ | $6.80 \times 10^4$ | 7.0 | 93 | 0.08 | $9.03 \times 10^5$ | | |
| Mn$^{II}$ | Cu$^{II}$ | $6.80 \times 10^5$ | 34.1 | 65.9 | 0.52 | $1.31 \times 10^6$ | $1.9(\pm 1.4) \times 10^6$ | −79.1 |
| Mn$^{II}$ | Cu$^{II}$ | $6.80 \times 10^6$ | 65.6 | 34.4 | 1.91 | $3.57 \times 10^6$ | | |
| Zn$^{II}$ | Cu$^{I}$ | $1.47 \times 10^0$ | 32.0 | 68.0 | 0.47 | $^d 2.49 \times 10^4$ | | |
| Zn$^{II}$ | Cu$^{I}$ | $8.18 \times 10^{-1}$ | 8.3 | 91.7 | 0.09 | $^d 7.22 \times 10^4$ | $4.0(\pm 2.8) \times 10^4$ | −69.5 |
| Zn$^{II}$ | Cu$^{I}$ | $1.15 \times 10^0$ | 29.5 | 70.5 | 0.42 | $^d 2.19 \times 10^4$ | | |

$^a$Ratio of metal availabilities buffered according to Supplementary Table 1 and Supplementary Data 1.

$^b$The proportion of each competing metal acquired by MncA as determined from the chromatograms in Fig. 1d.

$^c$Preference for M2 is calculated as the ratio of buffered concentrations of M1 and M2 (3$^{rd}$ column) divided by the ratios of occupancies of Metal1 and Metal2 (6$^{th}$ column) except where noted.

$^d$Calculated for M2 (Cu$^{I}$) relative to Mn$^{II}$ based on the observed preference of Zn$^{II}$ relative to Mn$^{II}$.

$^e$For M2 relative to an assigned value of −43.3 kJ mol$^{-1}$ for Mn$^{II}$ (defined in the legend to Fig. 2e and elaborated in Supplementary Figs. 5a, b). As pseudo-dissociation constants: Mn$^{II}$ $2.6 \times 10^{-8}$ M; Fe$^{II}$ $1.23 \times 10^{-9}$ M; Co$^{II}$ $1.78 \times 10^{-10}$ M; Ni$^{II}$ $3.61 \times 10^{-12}$ M; Zn$^{II}$ $3.26 \times 10^{-12}$ M; Cu$^{I}$ $6.58 \times 10^{-13}$ M; Cu$^{II}$ $1.37 \times 10^{-14}$ M.

$^f$A fourth trial competing Mn$^{II}$ with Ni$^{II}$ is not shown in Fig. 1c and the chromatograms are in Supplemental Fig. 2b. This result is in the calculated preference for Ni$^{II}$ (where $n = 4$ experimental replicates, $n = 3$ for the other determinations ± SD).

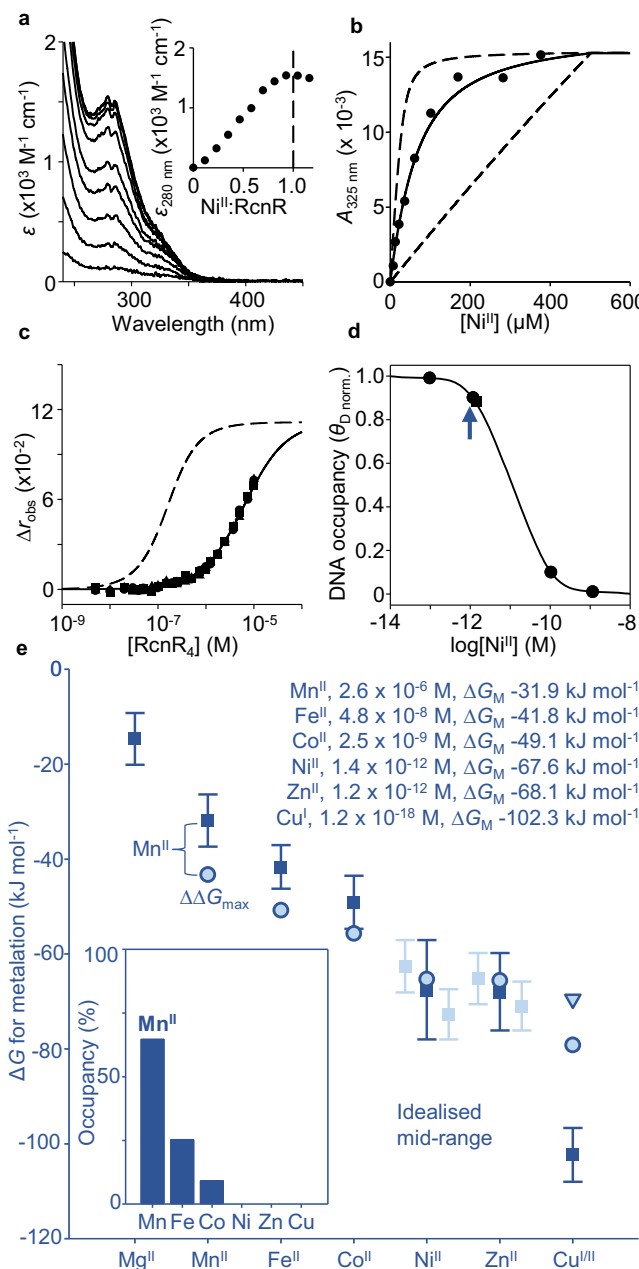

**Fig. 2 | Ni$^{II}$-RcnR refined mid-range metal availabilities decode Mn$^{II}$ as the cognate MncA metal. a** Apo-subtracted difference spectra of RcnR (17.2 μM RcnR monomer, ε calculated from total protein) titrated with Ni$^{II}$, inset showing peak wavelength confirming 1:1 stoichiometry of Ni$^{II}$ to RcnR monomer, or 4:1 to RcnR$_4$ ($n$ = 1). RcnR (20 μM monomer) also migrated with one equivalent of N$^{III}$ by SEC (Supplementary Fig. 3b). **b** Representative Ni$^{II}$ titration of RcnR (31.5 μM monomer) in EGTA (464 μM), solid line representing calculated $K_D$ for Ni$^{II}$ from simultaneously fitted RncR-EGTA competitions ($n$ = 4 independent experiments) at varied EGTA concentrations (Supplementary Fig. 3c, fitting models in Supplementary Software). Dashed lines, simulations with affinities 10-fold tighter and weaker than calculated $K_D$. **c** RcnR binding to hexachlorofluorescein-labelled *rcnRA* operator-promoter (10 nM) by fluorescence anisotropy. Solid line, best simultaneous fit to $n$ = 3 experimental replicates (circles, triangles, squares) for Ni$^{II}$-RcnR (fitting model in Supplementary Software), dashed line simulates apo-RcnR using published $K_D$ 1.5 × 10$^{-7}$ M and maximum $\Delta r_{obs}$ 0.1115[39]. **d** Ni$^{II}$ and DNA affinities determined above used with previously measured RcnR molecules cell$^{-1}$ to calculate (via Supplementary Data 2) relationship between intracellular Ni$^{II}$ availability and RcnR DNA occupancy ($\theta_D$), as for Co$^{II}$ (circles show $\theta_D$ 0.99, 0.90, 0.10 and 0.01). Combined mid-point of ranges for Ni$^{II}$-RcnR and Ni$^{II}$-NikR also shown (blue arrow). **e** Metal availabilities (squares and inset text) as activities/concentrations and free energies, $\Delta G_M$, at mid-points (50% DNA occupancies, representing idealised cells) of metal sensors (bars are sensor ranges, 10% to 90%), now including Ni$^{II}$-RcnR. Pale bars show individual ranges where two cognate sensors. MncA metal preferences (pale blue circles, Cu$^I$ triangles, from Fig. 1d) as free energies ($\Delta G_{MP}$) from pseudo-affinities giving 99% Mn$^{II}$ metalation at mid-range Mn$^{II}$ availability ($\Delta G_M$) without competing metals (Supplementary Fig. 5 simulates alternative values for Mn$^{II}$-MncA $\Delta G_{MP}$). Inset shows occupancies of MncA predicted from free energy differences between MncA and labile metal ($\Delta\Delta G = \Delta G_{MP} - \Delta G_M$). Mn$^{II}$ has the largest favourable gradient annotated $\Delta\Delta G_{max}$. Supplementary Data 3 enables similar predictions of cognate metals for other proteins. Source data are provided as a Source Data file.

of RcnR Ni$^{II}$ affinity 2.36 (± 0.13) × 10$^{-12}$ M from a simultaneous fit to $n$ = 4 titrations in different EGTA concentrations (Supplementary Fig. 3c). A representative data set confirms the fitted value is within the limits of the assay from dashed lines simulating values ten times tighter and weaker (Fig. 2b). Binding of apo- and of Co$^{II}$-RcnR to DNA was previously monitored by fluorescence anisotropy using hexachlorofluorescein-labelled *rcnA* operator-promoter fragments, to determine DNA affinities[39]. Here analogous titrations confirm that Ni$^{II}$ similarly weakens DNA-binding with fitted affinity of 3.09 (± 0.04) × 10$^{-6}$ M (Fig. 2c). Using these values, with previous apo-RcnR DNA affinity of 1.5 × 10$^{-7}$ M, plus known RcnR molecules cell$^{-1}$, the relationship between intracellular Ni$^{II}$ availability and *rcnA* operator-promoter occupancy was calculated as for Co$^{II}$ (Supplementary Data 2)[23]. *rcnA* transcripts increase as RcnR DNA occupancy decreases with elevated intracellular Ni$^{II}$ following the relationship in Fig. 2d. Metal availabilities at DNA occupancies of 0.99, 0.90, 0.10 and 0.01 are indicated. Analogous relationships for other metal sensors are in Supplementary Fig. 4. The bars on Fig. 2e show ranges of availability (as $\Delta G_M$) corresponding to occupancies of 0.1–0.9[23]. For Ni$^{II}$ the range in Fig. 2e now combines those for Ni$^{II}$-NikR and Ni$^{II}$-RcnR. The mid-point of the combined range is shown along with the mid-points for other metals, annotated as concentrations (M) and free energies (kJ mol$^{-1}$) (Fig. 2e). The Irving-Williams series is ambiguous about the order of Zn$^{II}$ versus Ni$^{II}$, but both are weaker than copper (Fig. 1a). Including Ni$^{II}$-RcnR, Fig. 2e reverses the order of intracellular availabilities of Zn$^{II}$ and Ni$^{II}$ shown in previous iterations[23], but still the sensors maintain availabilities to the inverse of the Irving-Williams series[2].

## Ni$^{II}$-RcnR refined mid-range metal availabilities decode correct metalation

Metalation within a cell should be predictable from the binding preferences of MncA in Fig. 1d relative to how tightly available (exchangeable) intracellular metals are bound, for example at the mid-points of sensor ranges (Fig. 2e). To make this comparison, Mn$^{II}$-MncA was assigned a pseudo-affinity (2.6 x 10$^{-8}$ M) giving 99% Mn$^{II}$-metalation

## Ni$^{II}$ availabilities defined by Ni$^{II}$-RcnR refine mid-range metal availabilities

Figure 1d presents the metal preferences of MncA after accounting for competition from other ligands (NTA or histidine). The speciation of protein metalation in the crowded cytosol is similarly thought to result from competition with diverse ligands binding labile, exchangeable metals at different availabilities[23,30–38]. Metal availabilities in the cytosol of bacterial cells (*Salmonella* and *E. coli*) have been estimated as free energies ($\Delta G_M$): formally free energies for complex formation with a notional half-metalated ligand at the respective availability[22,23]. DNA-binding metal sensors detect changes in $\Delta G_M$ and their responses have been calibrated for these bacteria[22,23,39]. However, the Ni$^{II}$ responses of high Ni$^{II}$-sensing RcnR were previously overlooked, and its calibration requires determination of Ni$^{II}$ affinity plus DNA affinity of Ni$^{II}$-RcnR (Supplementary Fig. 3a)[23,24,40,41].

RcnR absorbance changes with Ni$^{II}$ and a difference spectrum saturates at one equivalent per monomer, or 4:1 Ni$^{II}$:RcnR$_4$ (Fig. 2a). RcnR co-elutes with one Ni$^{II}$ atom per monomer by SEC (Supplementary Fig. 3b). EGTA competes for Ni$^{II}$ with RcnR enabling determination

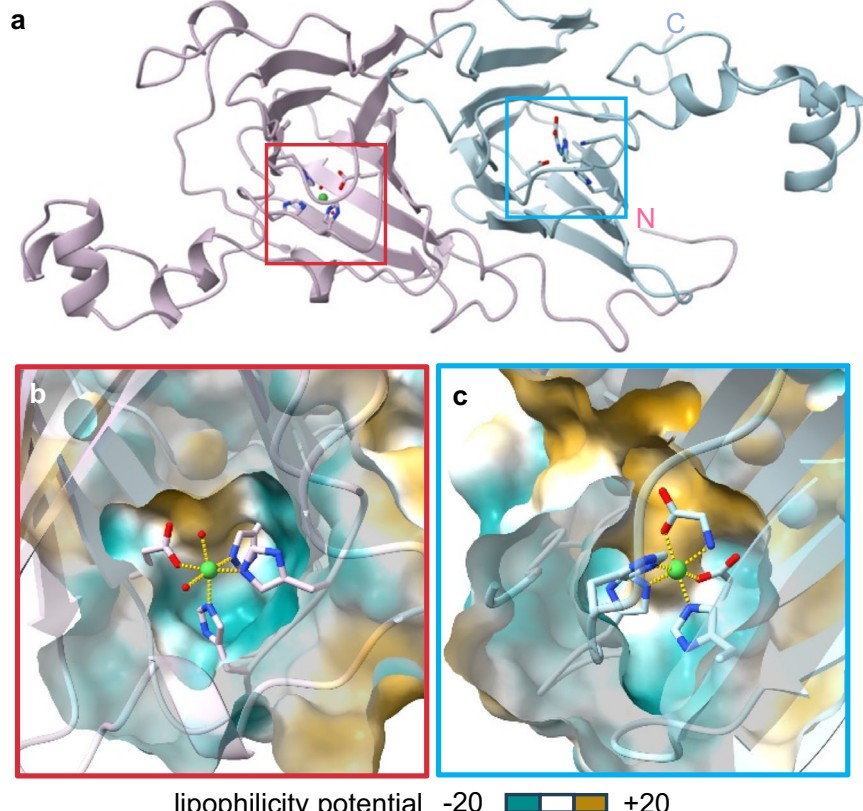

**Fig. 3 | Ni$^{II}$-MncA structure indicates non-cognate metal is also kinetically trapped. a** Ribbon representation of crystal structure of (Ni$^{II}$)$_2$MncA at 1.6 Å resolution showing characteristic bi-cupin fold and metal sites (boxed) modelled from residue 39 of the full protein sequence including signal peptide which is absent from the expressed protein (data collection and refinement statistics shown in Supplementary Table 3). Amino-terminal domain (pink), carboxy-terminal domain (blue) from residue 238 (in the full sequence). **b, c** Cross sections of solvent accessibility surfaces (modelled at 1.1 Å solvent radius to encompass dynamics) surrounding the metal sites show no channel to amino-terminal Ni$^{II}$ and narrow lipophilic (hydrophobic) channel to carboxy-terminal Ni$^{II}$. Ni$^{II}$ becomes trapped in the folded protein in a near octahedral geometry (yellow bonds; Supplementary Fig. 6) analogous to cognate Mn$^{II}$. The MncA model illustrates kinetic trapping of non-cognate metals suggesting MncA may be used to faithfully report in-cell metalation. Supplementary Fig. 7 shows that post-folding Ni$^{II}$ did not exchange with Cu$^{II}$ in vitro.

at the $\Delta G_M$ mid-point for Mn$^{II}$, and free energies of metal-MncA complex formation were designated $\Delta G_{MP}$. Values ($\Delta G_{MP}$) for other metals were then calculated using Fig. 1d (Table 1). The gradients from exchangeable cytosolic sites to MncA ($\Delta\Delta G$, formally $\Delta G_{MP}$ - $\Delta G_M$)[22,23], were calculated and inter-metal competition accounted for using the Ni$^{II}$-RcnR-refined calculator in Supplementary Data 3, to predict occupancies (Fig. 2e). The cognate metal Mn$^{II}$ was correctly decoded (largest favourable $\Delta\Delta G$). Pseudo-affinities ten times tighter or weaker generated the same proportional occupancies (Supplementary Fig. 5a, b). Thus, the speciation of metalation is a function of relative metal-binding preferences and availabilities, but total metal occupancy does vary with absolute $\Delta G_{MP}$ values for Mn$^{II}$-MncA. Using the Ni$^{II}$-RcnR-revised mid-point availabilities with four exemplar proteins whose affinities have been measured[22,23], decodes their cognate metals despite all preferring copper (Supplementary Fig. 5c–f). The Ni$^{II}$-RcnR revised calculator in Supplementary Data 3 can be used to decode cognate metalation.

### Non-cognate metal is also kinetically trapped by MncA

The labile character of protein-bound metals creates a challenge to define the in-cell metalation states of metalloproteins. Metals can be lost, gained, or exchanged at cell lysis and/or during purification and analysis. MncA is attractive because Mn$^{II}$ becomes trapped in the folded protein. However, MncA might adopt non-native folds with non-cognate metals. Abnormally coordinated metals might not be kinetically trapped. Ni$^{II}$ often prefers four-coordinate, planar geometries, and Ni$^{II}$-MncA was chosen for structural analysis. A crystal structure of Ni$^{II}$-MncA (1.6 Å resolution, Supplementary Table 3, Supplementary Fig. 6a), shows the bi-cupin fold of Mn$^{II}$-MncA with a metal atom in each cupin domain (Fig. 3a)[8]. Each Mn$^{II}$ atom is coordinated to three histidine and one glutamate residue with non-protein ligands completing hexacoordinate Mn$^{II}$ coordination spheres. Ni$^{II}$-MncA has similar coordination environments with no channel to the amino-terminal Ni$^{II}$ and a narrow channel to the carboxy-terminal site analogous to Mn$^{II}$-MncA (Fig. 3b, c)[8]. The channel is presumed to allow substrate access to catalytic Mn$^{II}$. The substrate site was gratuitously occupied by acetate in the Mn$^{II}$ structure and with glycine in the crystalised Ni$^{II}$-form (Supplementary Fig. 6, Fig. 3c). The narrow hydrophobic channel is unlikely to allow Ni$^{II}$ exchange and thus both Ni$^{II}$ atoms appear trapped.

MncA has >100-fold preference for Cu$^{II}$ over Ni$^{II}$ at folding (Fig. 1d). To test if Ni$^{II}$ is trapped, Ni$^{II}$-MncA was incubated for 24 h in a two-fold molar excess of Cu$^{II}$, bound and free metal separated by SEC followed by ICP-MS (Supplementary Fig. 7). The protein remained exclusively bound to Ni$^{II}$ confirming Ni$^{II}$ is kinetically trapped.

### Mis-metalation of MncA with Fe$^{II}$ predicted and observed in *E. coli*

Figure 4 (unlike Fig. 2e) shows estimated metal availabilities in actual *E. coli* BW25113 (elsewhere *E. coli*) grown aerobically[27], and the metalation

of soluble MncA in *E. coli*. Transcripts regulated by metal-sensors were previously quantified by qPCR, then related to promoter occupancies and hence metal availabilities via the relationships in Supplementary Fig. 4[27]. Notably, intracellular Ni[II] availability in aerobically grown cells is below the range for Ni[II]-RcnR. The largest favourable gradient from available exchangeable metal to MncA, $\Delta\Delta G$, is for Fe[II] not cognate Mn[II] (Fig. 4c).

High-level protein expression in heterologous host cells can deplete cofactors, and likely contributed to formation of MncA-containing inclusion bodies (Fig. 1b). To monitor in-cell metalation, MncA was thus expressed at a low level from a tuneable promoter using low inducer, low temperature, and prolonged slow growth overnight in *E. coli* (Fig. 4a). MncA was enriched from a soluble protein extract via anion exchange and SEC followed by ICP-MS. Metalation was quantified in three biologically independent experiments (Supplementary Fig. 8, Supplementary Table 4a, b). Absorbance at 280 nm largely reports MncA concentration due to its high extinction coefficient ($\varepsilon = 120,000$ M$^{-1}$ cm$^{-1}$), evident from stoichiometry approximating 2:1, and suggesting no apo-protein (Supplementary Table 4a, b). Some metals might alter the extinction coefficient and traces of interfering proteins could introduce variation between experiments. Proportional occupancies have thus been calculated from total metal in one or more MncA-containing fraction(s) rather than MncA concentration, notably with similar outcomes when duplicated (Supplementary Table 4b). The experiment was repeated using a single anion exchange step and analytical HPLC SEC with similar outcome (Supplementary Table 4). The mean ± SD MncA-speciation is shown (inset Fig. 4c). Speciation accounting for inter-metal competition was calculated using Supplementary Data 4. Predicted Fe[II] mis-metalation closely matches observed mis-metalation in *E. coli* (Fig. 4b, c).

## Substantial cognate MncA metalation predicted and observed in 4 mM manganese

We were eager to know if intracellular Mn[II] could be increased to a tolerable availability inside viable *E. coli*, sufficient to predominantly form Mn[II]-MncA. Fig. 4c shows metal preferences ($\Delta G_{MP}$) following a similar trend to availabilities ($\Delta G_M$), predicting that modest changes could switch the speciation of metalation. Using only manganese supplementation maximum intracellular Mn[II] availability was detected in media plus 4 mM manganese[27]. Notably, high manganese and hydrogen peroxide defined the 0.99 sensor boundary (Supplementary Fig. 4)[27], because manganese import is modulated by OxyR[42,43]. Here, 4 mM manganese did not inhibit cell density after prolonged culturing (Supplementary Fig. 9). A switch to predominant Mn[II] metalation was observed in three independent *E. coli* cultures supplemented with 4 mM manganese (Fig. 5a, inset Fig. 5b, Supplementary Table 5a, b, Supplementary Fig. 10).

Calibrated *mntS* transcript abundance in 4 mM manganese reads out intracellular availability of $4.5 \times 10^{-5}$ M ($\Delta G_M$ −24.8 kJ mol$^{-1}$) (Supplementary Fig. 4)[27]. This value is shown in Fig. 5b plus availabilities for other metals in un-supplemented media. The largest favourable free energy gradient from exchangeable available metals to MncA becomes Mn[II] (Fig. 5b). Metalation was predicted by substituting this elevated Mn[II] availability into Supplementary Data 4 (inset Fig. 5b). A switch to predominant metalation with cognate Mn[II] plus partial mis-metalation with Fe[II] is thus predicted, as well as observed by MncA-trapping, in *E. coli* supplemented with 4 mM manganese.

## MncA-trapped metals confirm negligible Zn[II] or Cu[I] metalation in supplemented cells

We further wondered if MncA could be predictably metalated with other elements in metal-supplemented viable *E. coli*. Availabilities of Zn[II] and Cu[I] at the upper (0.99) sensor boundaries (Supplementary

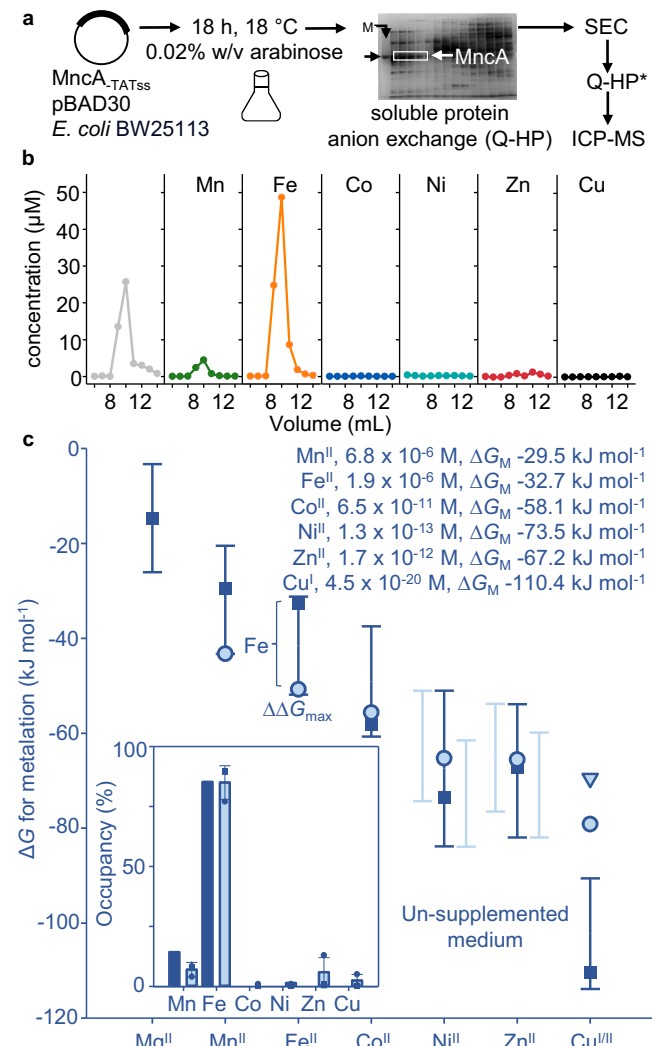

**Fig. 4 | Mis-metalation with Fe[II] predicted and observed in heterologous *E. coli*. a** Purification of soluble MncA folded in vivo, minus secretion signal, by anion exchange chromatography (5 mL Q-Sepharose, and 1 mL Q-Sepharose) both eluted using high-salt buffer, with intervening SEC (Superdex 200 or 75). MncA was recovered from a soluble protein fraction of *E. coli* after low-level expression overnight at 18 °C using pBAD30·*mncA*, induced with 0.02% w/v arabinose. Representative of *n* = 18 biologically independent purifications. Full gel image in Supplementary Fig. 8a. **b** Representative (*n* = 3 independent biological replicates) chromatogram showing in-cell acquired metals in MncA-containing fractions determined by ICP-MS revealing mis-metalation with Fe[II] and traces of Mn[II]. MncA-containing fractions were confirmed by SDS-PAGE (Supplementary Fig. 8d) and quantified by $A_{280nm}$ (using $\varepsilon = 120,000$ M$^{-1}$). Metal contents of MncA-containing fractions (Supplementary Table 4a, b, Supplementary Fig. 8) were used to calculate fractional occupancies (%). Supplementary Table 4 shows similar outcomes via a modified protocol as used in subsequent experiments. **c** Metal availabilities ($\Delta G_M$, squares and text inset) in the cytosol of *E. coli* grown aerobically in LB, estimated as in Foster et al. [27] by calibrated qPCR with genes regulated by cognate metal sensors (bars are sensor ranges from 1% to 99%). MncA metal-preferences ($\Delta G_{MP}$, pale blue circles), Cu[I] (triangle). Inset shows occupancies of MncA predicted from the free energy differences using Supplementary Data 4. The inset shows the resulting predicted MncA mis-metalation with Fe[II] (dark blue columns) in the heterologous host based on the largest favourable gradient ($\Delta\Delta G_{max}$ in main figure). Inset also shows mean (± SD) in-cell metalation from the *n* = 3 biological replicates (pale blue columns, squares, circles, triangles, Supplementary Table 4b), closely matching predictions. Source data are provided as a Source Data file.

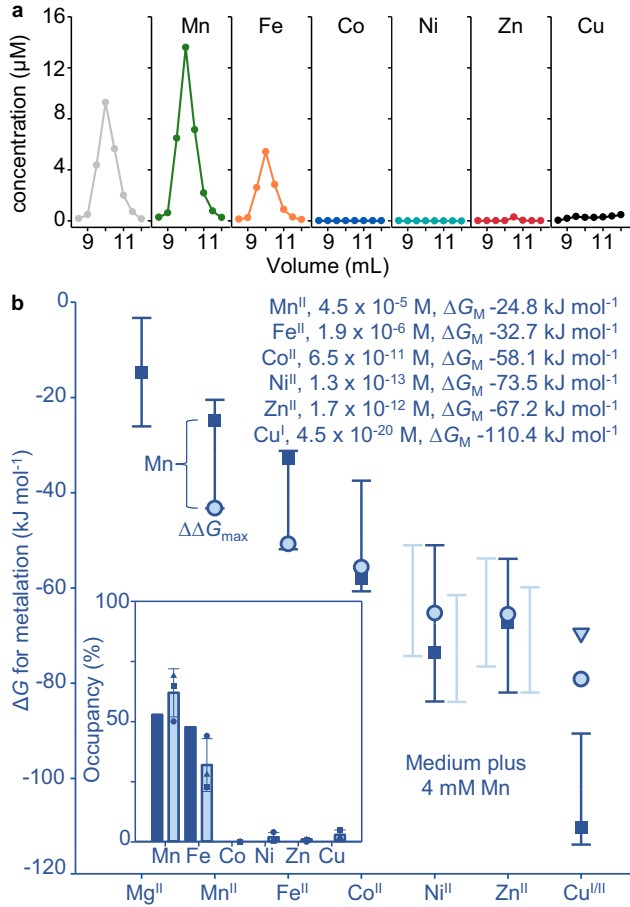

**Fig. 5 | Substantial cognate metalation predicted and confirmed in 4 mM manganese. a** Representative ($n = 3$ independent biological replicates) chromatogram showing in-cell acquired metals in MncA-containing fractions determined by ICP-MS, as in Fig. 4a, b but using a single anion exchange step (Q-Sepharose), HPLC SEC (TSK SW3000) and showing increased cognate metalation with Mn$^{II}$. MncA was identified by SDS-PAGE in Supplementary Fig. 10c. Metal contents of MncA-containing fractions (Supplementary Table 5) from independent biological replicates (Supplementary Fig. 10) were used to calculate fractional occupancies (%). **b** Metal availabilities ($\Delta G_M$, squares, and text inset) in *E. coli* cytosol grown aerobically as in Fig. 4c, except Mn$^{II}$ replaced with estimates from calibrated qPCR of MntR target *mntS* in cells cultured in 4 mM manganese (Supplementary Fig. 4). MncA metal-preferences as $\Delta G_{MP}$ (pale blue circles, Cu$^I$, triangle). Bars are sensor ranges from 1% to 99%. Inset shows occupancies of MncA predicted from free energy gradients using Supplementary Data 4 and substituting Mn$^{II}$ availability with a value of $4.5 \times 10^{-5}$ M (as in inset text). The inset shows predicted MncA metalation (dark blue columns) with Mn$^{II}$, based on the largest favourable gradient ($\Delta\Delta G_{max}$ in main figure), plus partial metalation with Fe$^{II}$. Inset also shows mean ($\pm$ SD) in-cell metalation from the $n = 3$ independent biological replicates (square, circle, triangle, pale blue columns, as in Supplementary Table 5), largely matching predictions. Source data are provided as a Source Data file.

Fig. 4)[23], annotated in Fig. 6a, were entered into the metalation-calculator in Supplementary Data 4 as before. Negligible or no occupancy with either metal, but Fe$^{II}$ mis-metalation, was predicted (Fig. 6a).

Predicted mis-metalation with Fe$^{II}$ (negligible Zn$^{II}$) was observed in cells grown in 800 µM zinc (Fig. 6a). This treatment was selected to approximate maximal abundance of *zntA* transcripts regulated by ZntR[27], hence maximum Zn$^{II}$ availability with slight inhibition of growth (Supplementary Fig. 9). Chromatographic profiles for Zn$^{II}$ (unlike Fe$^{II}$ and Mn$^{II}$) imperfectly align with absorbance at 280 nm or the distribution of MncA on SDS-PAGE (Supplementary Fig. 11). We did not

identify the contaminating Zn$^{II}$-protein by principal component analysis and MncA-metalation was determined from a single fraction showing least evidence of other proteins (Supplementary Fig. 11, Supplementary Table 6).

Predicted mis-metalation with Fe$^{II}$ (not Cu$^I$) was also observed in cells grown in 600 µM copper (Fig. 6a, Supplementary Fig. 9). The profiles for copper (unlike Fe$^{II}$ and Mn$^{II}$) again imperfectly align with MncA but correlated with a protein of $M_r$ matching glyceraldehyde-3-phosphate dehydrogenase (GAPDH) (Supplementary Fig. 12). GAPDH is known to bind copper in copper-exposed cells and is removable using Blue Sepharose[44]. Including this step retained Fe$^{II}$-MncA and Mn$^{II}$-MncA but eliminated co-migrating copper (Supplementary Fig. 13). Metalation was quantified from three independent cultures after using Blue Sepharose (Supplementary Table 7a, b). In summary, intracellular availabilities of Cu$^I$ or Zn$^{II}$ are insufficient to metalate MncA in supplemented cells, with metalation matching predictions.

### MncA-trapped metals in high Ni$^{II}$ and Co$^{II}$ refine availabilities

MncA trapped Ni$^{II}$ or Co$^{II}$ in the respective metal supplemented media (Fig. 6a, Supplementary Fig. 9, 14, 15, Supplementary Table 8,9). Metalation was predicted as for high Zn$^{II}$ and Cu$^I$ but using 0.01 boundary values for RcnR-DNA occupancy (Supplementary Fig. 4). In high cobalt, MncA bound more Co$^{II}$, less Fe$^{II}$ and more Mn$^{II}$, than predicted (Fig. 6a). The most parsimonious explanation is that Fe$^{II}$ availability declines in high Co$^{II}$. An MncA-residuals calculator in Supplementary Data 5, formulated in Supplementary Note 2, used the determined preferences of MncA to estimate the decrease in Fe$^{II}$ availability relative to Co$^{II}$ (Fig. 6b). Figure 6c compares observed occupancies with predictions using the refined Fe$^{II}$ availability. Residuals for Mn$^{II}$, not used in refinement, reduced as anticipated for the parsimonious solution.

In high Ni$^{II}$, MncA bound more Ni$^{II}$, less Mn$^{II}$ and Fe$^{II}$, than predicted (Fig. 6a). Here, the most parsimonious explanation is that Ni$^{II}$ is more available than approximated. The MncA-residuals calculator estimated the further increase in intracellular Ni$^{II}$ availability relative to Fe$^{II}$ (Fig. 6b). Figure. 6c compares Ni$^{II}$-refined occupancies with those observed. Residuals for Mn$^{II}$ again reduced. In summary, MncA metal-occupancies plus the MncA-residuals calculator can be used to probe and refine estimates of intracellular metal availabilities. These tools can assist predictions of in-cell metalation of proteins in other cell types and growth conditions. MncA-derived refinements also suggest Fe$^{II}$ availability declines in *E. coli* in high cobalt and this is subsequently explored.

### Shallow Mn$^{II}$ pool is depleted in un-supplemented media

Although Mn$^{II}$ is a most available metal (least negative $\Delta G_M$, Fig. 4c), the Mn$^{II}$ pool in un-supplemented media is shallow, containing only a few thousand atoms per cell[26]. qPCR with primers to MntR-regulated *mntS* revealed increased expression and hence reduced available Mn$^{II}$ (Mn$^{II}$-MntR co-repression being alleviated) during prolonged MncA expression in un-supplemented medium (Fig. 7a). This is consistent with observing less Mn$^{II}$-MncA than predicted (Fig. 4c inset). Fur-regulated *fepD* transcripts indicate a smaller decline in the available Fe$^{II}$ pool in un-supplemented medium. These data also highlight a challenge with qPCR for estimating intracellular metal availabilities at upper transcript abundance, corresponding to lower availabilities for metal-dependent co-repressors MntR and Fur. The logarithmic nature of PCR as represented by the log$_2$ scales in Fig. 7, means the upper 50% of the expression range corresponds to a single PCR cycle. The abundance of *mntS* transcripts reaches (and exceeds) the upper boundary, lowest available Mn$^{II}$, reported previously whereas *fepD* remains 3 to 4 cycles below (Fig. 7a, b)[27]. These data thus report depletion of the intracellular Mn$^{II}$ pool overnight by ~18 h but a relatively modest change in Fe$^{II}$. The use of MncA to refine estimated availabilities is thus especially valuable for qPCR-based estimates in the

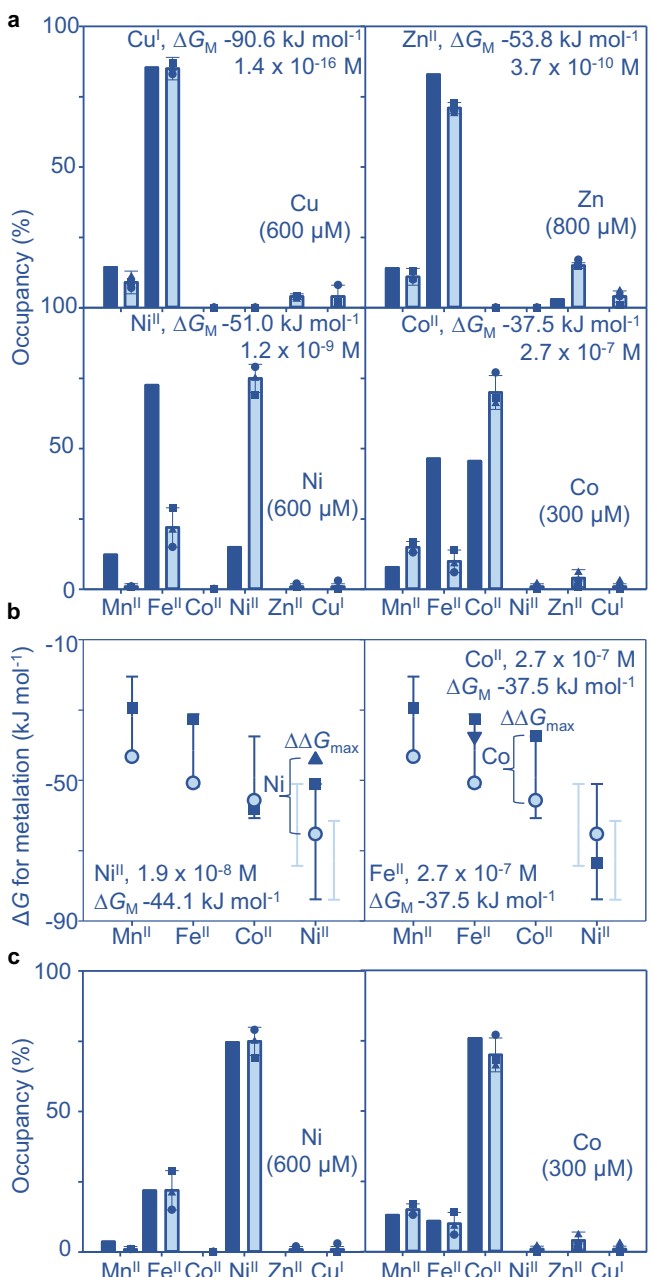

**Fig. 6 | MncA read-outs of in-cell metalation in high $Ni^{II}$ and $Co^{II}$ refine metal availabilities. a** In-cell metalation with $Ni^{II}$ and $Co^{II}$, not $Cu^{I}$ or $Zn^{II}$, switches from $Fe^{II}$-MncA in metal-supplemented (600 μM, 300 μM, 600 μM, 800 μM respectively as shown) media, qualitatively matching predictions but quantitatively greater $Ni^{II}$ and $Co^{II}$ metalation than predicted. Metalation (dark blue bars) predicted using Supplementary Data 4 from availabilities in un-supplemented medium but with separately altered high availabilities of $Cu^{I}$, $Zn^{II}$, $Ni^{II}$ or $Co^{II}$ (as inset text). High availabilities correspond to the respective upper metal boundaries ($\theta_{DM}$ or $\theta_{D}$, 0.99 or 0.01) shown in Supplementary Fig. 4. Mean ($\pm$ SD) measurements of in-cell metalation ($n = 3$ independent biological replicates, square, circle, triangle) in high metal (pale blue bars) calculated using data in Supplementary Tables 6–9 based on Supplementary Figs. 11–15. **b** Selected intracellular metal availabilities ($\Delta G_M$) refined (triangles) using Supplementary Data 5 for $Ni^{II}$ (left) from observed in-cell $Fe^{II}$- and $Ni^{II}$-MncA occupancies in high $Ni^{II}$ (relative to $Fe^{II}$ in un-supplemented medium), and for $Fe^{II}$ (right) from $Co^{II}$ and $Fe^{II}$ occupancies in $Co^{II}$ (relative to elevated $Co^{II}$ as show in inset text). Metal preferences of MncA shown as $\Delta G_{MP}$ (circles). Bars are sensor ranges from 1% to 99%. **c** Calculated metalation (dark blue bars) using Supplementary Data 4 but with refined availabilities as in panel (**b**), reduced residuals for $Mn^{II}$ (which was not included in the refinement process) relative to observed in-cell metalation (pale blue bars included for comparison), showing mean ($\pm$ SD) ($n = 3$ independent biological replicates, square, circle, triangle). MncA metalation can be used to refine relative intracellular metal availabilities and Supplementary Data 5 is provided to enable such calculations. Source data are provided as a Source Data file.

(Fig. 7a, c–f). This is also true for cultures that are not expressing MncA excluding the formal possibility that turnover of metalated MncA sustained elevated steady-state $Mn^{II}$, $Ni^{II}$ and $Co^{II}$ availabilities but not $Cu^{I}$ and $Zn^{II}$. Lack of steady-state elevated $Cu^{I}$ and $Zn^{II}$ indicates a difference in metallostasis for these metals compared to $Mn^{II}$, $Co^{II}$ and $Ni^{II}$. It may be more challenging to metalate proteins in viable cells with $Zn^{II}$ or $Cu^{I}$ simply by metal supplementation.

### Fur causes $Fe^{II}$ atoms cell$^{-1}$ to decline in $Co^{II}$

To investigate the MncA-predicted decline in intracellular $Fe^{II}$ availability in high $Co^{II}$ (Fig. 6b), total atoms of iron cell$^{-1}$ were measured by ICP-MS (Fig. 8a). Iron content declined in $Co^{II}$ consistent with the prediction (noting that available metal can sometimes trend differently to total metal). Cells lacking Fur contain less iron in standard media as explained previously[45]. Importantly, the effect of $Co^{II}$ on iron is absent in $\Delta fur$ (Fig. 8a). Fur binds $Co^{II}$ and $Co^{II}$-Fur binds DNA with both affinities already known[46]. Fig. 8b calculates the Fur response to intracellular $Co^{II}$ simulated as for $Fe^{II[23]}$. DNA occupancy at the $Co^{II}$ availability inside cells grown here in elevated cobalt reveals an extreme cross-response of Fur. Non-cognate sensors are known to cross-respond to availabilities at the top of the cognate sensor ranges[39]. Crucially, cross-metalation of Fur with $Co^{II}$ explains the decline in total iron in Fig. 8a and in available intracellular $Fe^{II}$ read out by MncA (Fig. 6b).

### MncA-trapped metals iteratively refine estimated availabilities

The remaining residual differences between observed and predicted occupancies in Fig. 6c have been used via the MncA-residuals calculator (Supplementary Data 5), to further refine intracellular metal availabilities (Supplementary Fig. 16). $Mn^{II}$ availability is inferred to decrease in high $Ni^{II}$ and increase in high $Co^{II}$. Both predicted trends were reflected in the total atoms cell$^{-1}$, and the effects of $Co^{II}$ on $Mn^{II}$ were not Fur-dependent (Fig. 8c). The effects of high $Ni^{II}$ on $Mn^{II}$ are not dependent on $Mn^{II}$-sensing MntR (Supplementary Fig. 17a). The MntR-independent decline in $Mn^{II}$ is less severe in wild-type where the sensor controls metallostasis dampening changes. The residuals in cells exposed to 4 mM manganese suggest a decline in available $Fe^{II}$ and this matched a Fur-independent decline in total iron atoms cell$^{-1}$ (Fig. 5b, Supplementary Fig. 17b). The MncA-residuals calculator was used iteratively to further refine high $Ni^{II}$, $Co^{II}$ and $Mn^{II}$ blueprints (Supplementary Fig. 16). The refined availabilities and related high-metal

upper 50% of expression ranges. This applies to some metal-supplemented cells but not un-supplemented cells as cultured previously[27]. Viewed together, Fig. 4c (inset) and Fig. 7a reveal that disparities between predicted and observed MncA-metalation can report on non-steady-state intracellular metal availabilities ($\Delta G_M$).

### Steady-state availabilities sustained for high $Mn^{II}$, $Ni^{II}$ and $Co^{II}$, not $Cu^{I}$ and $Zn^{II}$

At the outset of MncA expression, cells exposed to elevated metals all reach or exceed the estimated transcript abundances defined as the qPCR boundaries for high intracellular metal availabilities (Fig. 7). For $Mn^{II}$-MntR co-repressed *mntS* transcripts, this reflects high occupancy of the *mntS* promoter in cells exposed to 4 mM manganese and low transcript abundance. In contrast this equates to high transcript abundance for $Zn^{II}$, $Cu^{I}$, $Co^{II}$ and $Ni^{II}$ responsive transcripts regulated by activators ZntR, CueR and metal-dependent de-repressor RcnR. At the end of MncA production (18 h) in metal-supplemented cells, $Mn^{II}$, $Co^{II}$ and $Ni^{II}$ availabilities remain high, but they decline for $Zn^{II}$ and $Cu^{I}$

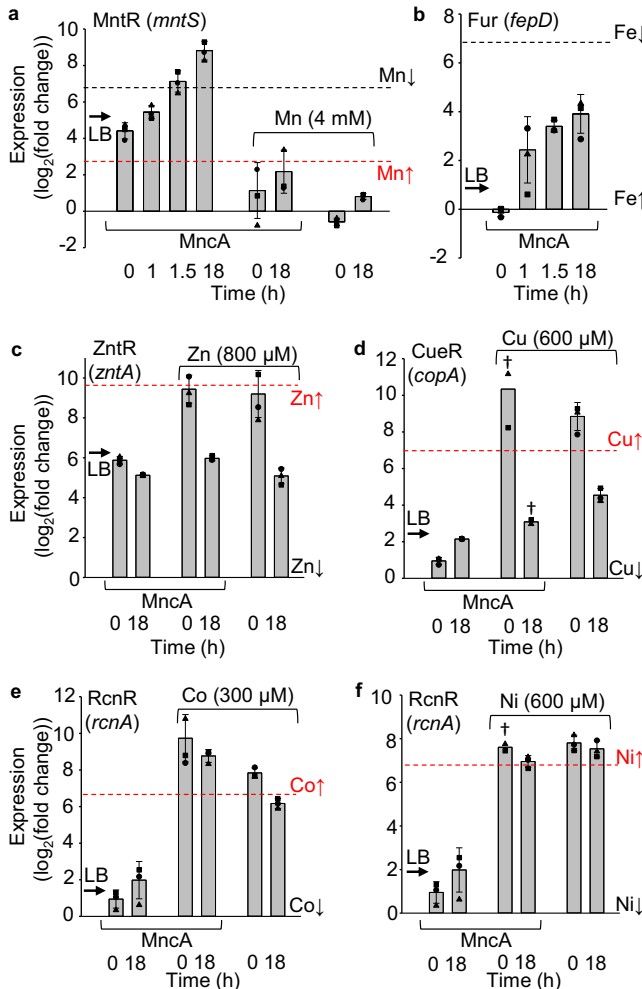

**Fig. 7 | Shallow Mn^II pool depleted in un-supplemented medium but steady-state availabilities sustained in high Mn^II, Ni^II and Co^II.** qPCR of **a** *mntS* (regulated by Mn^II-MntR), **b** *fepD* (Fe^II-Fur), **c** *zntA* (Zn^II-ZntR), **d** *copA* (Cu^I-CueR), **e** *rcnA* (Co^II-RcnR), **f** *rcnA* (Ni^II-RcnR), as log₂ change relative to respective lowest measured transcript abundance observed in Foster et al.[27] using chelator, metal and/or H₂O₂ supplemented cells ( - 0). ΔCq values in Source Data TXT for Fig. 7, data are mean ± SD of n = 3 independent biological replicates (square, circle, triangle) except where dagger n = 2. MntR and Fur target gene transcripts increase in low Mn^II and Fe^II (MntR and Fur are metal-dependent co-repressors), but other transcripts increase in high availabilities of cognate metals (regulated by activators or metal-dependent de-repressors). Previous log₂(fold change) qPCR at high metal-boundaries (red dashed lines, 0 for Fur) and at low metal boundaries (black dashed lines for MntR and Fur) are shown. Arrows show previous log₂(fold change) qPCR for cognate transcripts isolated from cells grown in LB[27]. Panels (**a**) and (**b**) show expression at 0 h, 1 h, 1.5 h, 18 h after addition of arabinose to un-supplemented medium to induce low expression of MncA at 18 °C. After 18 h MntR regulated *mntS* transcripts pass the low Mn^II boundary (DNA occupancy θ_D 0.01) (panel **a**) further quantified in Supplementary Fig. 18. The Mn^II pool is shallow and depleted in LB. For Fur-regulated *fepD* the log₂ scale reports modest depletion of intracellular available Fe^II at 18 h in un-supplemented medium (Supplementary Fig. 18). The first two columns (panels **c**–**f**) show negligible change in availability of Zn^II, Cu^I, Co^II or Ni^II after 18 h MncA expression in non-supplemented medium. Final four columns (panels **a**, **c**–**f**) report log₂(fold change) in metal-supplemented media either with or without expression of MncA as indicated. Steady-state Mn^II availability approximating upper Mn^II boundary sustained to 18 h in 4 mM manganese (panel **a**). Steady-state approximating the upper metal boundary is also sustained in high Ni^II and Co^II, but not Cu^I and Zn^II (panels **c**–**f**). Source data are provided as a Source Data file.

calculators are provided (Supplementary Data 6–8). MncA and the MncA-residuals calculator in Supplementary Data 5, could be used to probe relative metal availabilities in other cell types including (rating) cells with metallostasis engineered to optimise the speciation of metalation for selected elements.

## Discussion

Here we discover that the metal-binding preferences of a protein which kinetically traps metals at folding, MncA, all follow the Irving-Williams series (Fig. 1). This may reflect the preferences of flexible sites in the folding pathway prior to kinetic trapping. It is also possible that preferred metals in the series increasingly encourage progression of MncA folding intermediates. Although copper is most preferred, the cognate metal is correctly decoded to be Mn^II by relating these preferences to idealised estimates of intracellular metal availabilities, standardised to the mid-points of the ranges of metal sensors (Fig. 2). MncA kinetically traps metals, such that those which co-purify are likely to reflect in vivo metalation states (Fig. 3, Supplementary Fig. 7). Estimates of metal availabilities inside *E. coli* grown aerobically predict that (cyanobacterial) MncA will be mis-metalated with Fe^II in *E. coli*, and this is established by ICP-MS analysis of purified soluble MncA (Fig. 4). Thus, the speciation of metalation is directly shown to be determined by relative binding preferences for metals competing at intracellular availabilities (Supplementary Fig. 5a), and mis-metalation can be predicted in heterologous hosts. Subtle changes in a metal availability (or preference) can switch metalation because intracellular availabilities follow the inverse of the Irving-Williams series and hence trend with preferences (Figs. 2c, 4c, 5b, 6a). Estimates of intracellular metal-availabilities can thus be used to predict the speciation of in vivo metalation to inform the engineering of natural and synthetic metalloproteins to optimise metalation (using Supplementary Data 3, 4, 6–8).

Unrefined metal availabilities derived from qPCR-based responses of metal-sensing transcriptional regulators correctly predicted Fe^II mis-metalation of MncA in *E. coli* in un-supplemented media (Fig. 4c). In metal-supplemented media, MncA occupancies with Co^II, Ni^II and Mn^II exceeded (at least to some extent) predictions (Figs. 5, 6a). These occupancies were used with Supplementary Data 5 to refine estimates of intracellular availabilities predicting decreased available Fe^II in Co^II and Mn^II, decreased available Mn^II in Ni^II, but increased available Mn^II in Co^II (Fig. 6b, Supplementary Fig. 16). Encouragingly, in every case this coincided with analogous changes in total Fe^II or Mn^II atoms cell⁻¹ (Fig. 8a, c, Supplementary Fig. 17a, b). The decline in cellular Fe^II in high Co^II, but not in high Mn^II, was Fur-dependent and the former explained by responses to Co^II of Fur (Fig. 8a, b, Supplementary Fig. 17b). The extent to which cross-metalation of Fur with Co^II is disadvantageous mis-metalation, remains to be established. Together these data show the value of MncA as a probe of intracellular metal-availabilities, orthogonal to synthetic metal sensors[47–49], and complementing calibrated endogenous metal sensors especially where qPCR is less reliable and/or crosstalk likely[39].

MncA occupancies report lowered Mn^II availability inside cells after ~18 h growth in un-supplemented medium (Fig. 4). Here, the shallow Mn^II pool is depleted during MncA expression such that steady-state availability is not sustained (Fig. 7a, Supplementary Fig. 18). This is significant when recombinant metalloproteins are over-expressed in *E. coli* potentially depleting metals: metalation can then reflect depths of available metal pools rather than strengths of competition with other ligands. In manganese-, nickel- and cobalt-supplemented cells elevated intracellular availabilities were sustained for ~18 h but this was not the case in elevated zinc and copper (Fig. 7). Mechanisms of metallostasis for Zn^II and Cu^I must adjust rates of import, export and/or consumption to restore pre-exposure steady-state intracellular availabilities, whereas elevated steady-states are

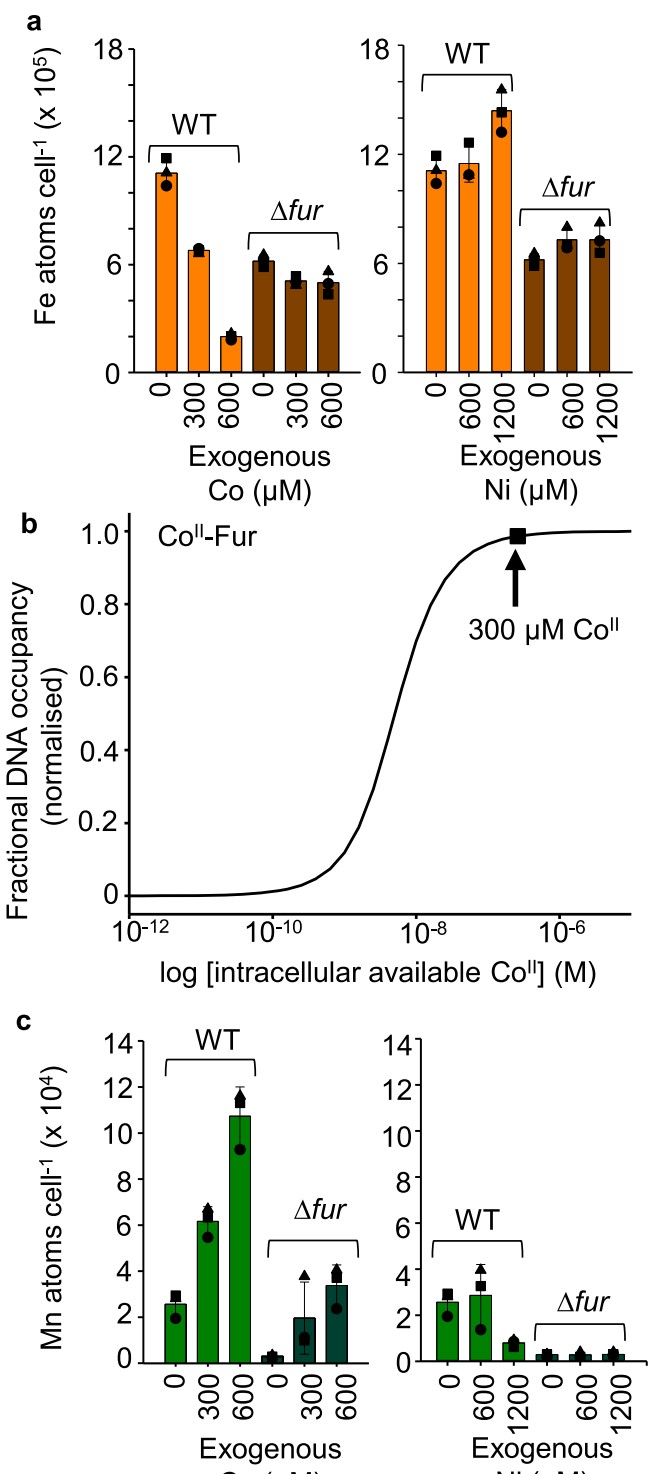

**Fig. 8 | Fur causes iron atoms cell⁻¹ to decline in CoII matching MncA-predicted availabilities. a** Total iron atoms cell⁻¹ determined by ICP-MS of cell digests, declines in cells grown in medium supplemented with high CoII (left panel). This is consistent with the decline in intracellular FeII availability estimated from the residuals in Fig. 6a, calculated using Supplementary Data 5 and shown in Fig. 6b. Elevated NiII does not affect total iron atoms cell⁻¹ (right panel). Total iron cell⁻¹ is less in Δ*fur* but does not further decline in elevated CoII. The decline in iron cell⁻¹ in response to elevated CoII is Fur-dependent. **b** Fur binds CoII to promote DNA-binding: simulated using known CoII affinity of Fur, DNA affinity of CoII-Fur, plus apo-Fur DNA affinity and Fur molecules cell⁻¹, as simulated for FeII-Fur[23]. Fur promoters will be aberrantly repressed in 300 μM cobalt (square and blue arrow) causing the predicted decline in intracellular available FeII shown in Fig. 6b and observed decline in total iron atoms cell⁻¹ in panel (**a**). **c** Total manganese atoms cell⁻¹ increase in high CoII consistent with slightly increased intracellular available MnII, modellable from the remaining residuals in high CoII in Fig. 6c, calculated using Supplementary Data 5, shown in Supplementary Fig. 16b. These effects of cobalt on MnII are independent of Fur. Total manganese atoms cell⁻¹ decrease in high NiII, this trend is more pronounced in Δ*mntR* (Supplementary Fig. 17), the trend is not evident in Δ*fur* which contains low manganese (right). An analogous decrease in available intracellular MnII calculable (Supplementary Data 5) from the remaining residuals in high NiII, Fig. 6b, is shown in Supplementary Fig. 16a. Use of MncA as a probe of relative intracellular metal availabilities iteratively refines values shown in Supplementary Fig. 16, included in Supplementary Data 6–8, and provided as blueprints to assist manipulation of the speciation of in vivo protein metalation. Mean ± SD of n = 3 independent biological replicates (squares, circles, triangles) in (**a**) and (**c**). Source data are provided as a Source Data file.

in vivo. Alternatively, mechanisms of metal homoeostasis could be exceptionally finely tuned to native metallo-proteomes minimising mis-metalation. In either eventuality, this raises the prospect in engineering biology of widespread mis-metalation because metal-availabilities will not necessary be correctly tuned to non-native metalloproteins. Notably, in *E. coli*, GTP-dependent metallochaperones YeiR and YjiA are predicted to be predominantly metalated with presumed cognate ZnII[22], whereas heterologous CobW from *Rhodobacter*, CbiK from *Salmonella*, in common with cyanobacterial MncA become mis-metalated with ZnII, FeII and FeII respectively (Fig. 4, Supplementary Fig. 20). Quantification of B₁₂ production in engineered *E. coli* provided indirect evidence of the predicted mis-metalation of CobW, and CbiK is known to insert FeII into siroheme in *Salmonella* missing the siroheme chelatase CysG[9,22]. Crucially, MncA now provides a direct read-out confirming predicted mis-metalation with FeII (Fig. 4).

MncA-refined intracellular metal availabilities along with the calculators provided here can guide optimisation of metalation via use of metal supplements or chelants, alterations to homeostasis by engineering host strains (chassis), or engineering proteins to match availabilities. Approximately half the reactions of life rely on the chemistries of the correct metals bound to metalloproteins[1]. Considerable research is being directed to the generation of various types of artificial metalloenzymes[29,50,51]. Inevitably, efforts in metabolic engineering, synthetic biology and directed evolution will often rely on metalloenzymes[52–54]. The blueprints and calculators can inform metalloenzyme engineered for in vivo bioprocessing applications. The optimisation of in vivo metalation presents opportunities as the engineering of biological systems for (sustainable) bio-manufacturing is prioritised[55].

## Methods

### Expression and purification of unfolded apo-MncA

To examine the relative binding preferences of MncA in vitro, protein was expressed and purified[8]. Briefly, MncA minus TAT secretion signal was expressed from pET29a-*mncA* in *E. coli* BL21(DE3) pLysS. Isopropyl β-D-1-thiogalactopyranoside (IPTG, 1 mL 0.4 M) was added to a mid-log phase culture (1 L in a 2 L flask) ~ OD₆₀₀ₙₘ 0.6–0.8 at 37 °C, to induce high-level expression for 3 h before harvesting cells by centrifugation

maintained for the other metals (Fig. 7). Nonetheless, it is predicted that MncA would bind negligible CuI or ZnII even if elevated steady-state availabilities were sustained. Notably, more favourable gradients (ΔΔG) for FeII and MnII largely exclude ZnII (Supplementary Fig. 19). Less FeII and MnII in the media, engineering cells to have reduced availabilities of FeII and MnII, or weakening MncA binding to FeII and MnII, could assist accumulation of ZnII-MncA.

Metalation of MncA inside *E. coli* switched between MnII, FeII, CoII, and NiII as predicted (Figs. 4–6). This becomes achievable because intracellular metal availabilities track with metal-binding preferences of proteins. This could indicate that protein mis-metalation is likely

(4000 × $g$, 4 °C) and freezing (−20 °C). The pellet was resuspended in 30 mL 100 mM Tris pH 7.5, 100 mM NaCl, 1 mM EDTA, 1 mM phenylmethylsulphonyl fluoride (PMSF) and sonicated (4 × 10 s on ice, 1 min intervals). Lysate was cleared by centrifugation (27,000× $g$, 15 min, 4 °C), supernatant discarded, and the pellet resuspended in 100 mM Tris pH 7.5, 100 mM NaCl, 1% (v/v) Triton X-100 (30 mL), sonicated (3 × 10 s on ice, 1 min intervals) and centrifugation repeated. The pellet was washed in 30 mL 100 mM Tris pH 7.5, 100 mM NaCl to remove the Triton, then sequentially in 30 mL 50 mM HEPES pH 7.5, 1 M urea followed by 15 mL 50 mM HEPES pH 7.5, 2 M urea. During the final wash the lysate was split into 2 mL aliquots followed by centrifugation (15,890× $g$, 10 min) to recover inclusion bodies, stored at −20 °C.

## Preparation of metal stocks

Metal stocks (MnCl$_2$, CoCl$_2$, NiSO$_4$, CuSO$_4$ and ZnSO$_4$) in ultrapure water were sterile filtered (0.2 µm) and quantified by inductively coupled plasma mass spectrometry (ICP-MS). When required for anaerobic experiments, (NH$_4$)$_2$Fe(SO$_4$)$_2$ stocks were prepared in N$_2$-purged ultrapure water. Total Mn and Fe concentrations were then confirmed by ICP-MS. Fe$^{II}$ stock was confirmed to be > 95% reduced by reaction with excess (10-fold) ferrozine (Fz) using $\varepsilon_{562nm} = 27,900$ cm$^{-1}$ M$^{-1}$ for the Fe$^{II}$Fz$_3$ complex. Reduced Cu$^{I}$ stocks were prepared as described in specific experiments and cuprous state validated with bicinchoninic acid (BCA) and ICP-MS.

## Production of buffered competing metals

MncA in vitro refolding buffers (in 50 mM MOPS pH 7.5) contained pairs of metals and ligand in varied amounts to achieve different buffered [available metals]. Buffers were prepared in acid-washed flasks with components added in the order, pH buffer, ultrapure water, ligand (typically NTA, or 1 mM L-histidine for competitions with Ni$^{II}$) followed by the two metals. Buffers were typically prepared in 100 mL volumes, filtered via 0.45 µm filters if any evidence of light scatter. Supplementary Table 1 shows the total amounts of each metal and ligand (NTA, histidine or neither) plus the buffered [available metals] used in each refolding solution. Supplementary Data 1 (derivation in Supplementary Note 1) was used and is provided here to assist in the production of such paired metal buffers. For an effective buffer the total metal concentration should substantially exceed the protein concentration ( >100-fold generally, 10-fold for competitions with Cu$^{I}$). To achieve an effective buffer < 80% of the buffering agent should be metalated.

A preliminary experiment was performed with BCA to buffer Cu$^{I}$ with no expected buffering of Mn$^{II}$. With 400 µM BCA and 50 µM Cu$^{I}$, expected free [Cu$^{I}$] is 10$^{-15}$ M, which was competed against 10 µM MnCl$_2$. Because < 1% copper co-migrated with MncA, unbuffered Zn$^{II}$ was competed against unbuffered Cu$^{I}$. Typical concentrations were 10 µM each and were verified by ICP-MS on the folding solution. Competitions between Zn$^{II}$ and Cu$^{I}$, used freshly prepared solution of CuCl (10 mM CuCl, 1 M NaCl, 0.1 M HCl) and refolding experiments included 1 mM hydroxylamine to maintain copper in reduced form. For buffers forming 2:1 metal-dependent complexes, buffered concentrations of each metal were determined using *HySS* software for His[56]. Log$\beta$ values for all equilibria are as follows: proton dissociation from histidine (HisH-9.08, HisH$_2$-15.1, HisH$_3$-16.8), His complexation of Mn$^{II}$ (Mn$^{II}$His-3.3, Mn$^{II}$His$_2$-6.3), His complexation of Ni$^{II}$ (Ni$^{II}$His-8.67, Ni$^{II}$His$_2$-15.54), p$K_w$ = 13.8[57]. For Cu$^{I}$ BCA complexes, derivation of buffering of [Cu$^{I}$] in Supplementary Note 1.

## Metal-binding preferences of MncA at folding

Inclusion bodies containing MncA were solubilized in HEPES pH 7.5 with 8 M urea, and the concentration calculated from $A_{280nm}$ using an experimentally determined extinction coefficient (120,000 cm$^{-1}$ M$^{-1}$)[8], typically in the range 100–500 µM. Refolding was achieved by

dropwise dilution of urea-solubilised MncA in large volumes (100 mL) of solutions containing pairs of competing buffered metals prepared as described earlier. Solutions were thoroughly mixed before adding unfolded MncA and gently mixed between additions. Dilute refolded MncA was recovered by binding to a 1 mL Q-Sepharose (Cytiva or GE Healthcare) anion exchange column pre-equilibrated with low-salt buffer (50 mM Tris pH 7.5, 50 mM NaCl). The column was washed with 20 mL low-salt buffer before eluting MncA with high-salt buffer, 50 mM Tris pH 7.5, 500 mM NaCl. MncA was quantified by $A_{280nm}$ then resolved (0.5 mL at -10 µM) by SEC (PD-10, GE healthcare, previously washed with 0.5 mL 5 mM EDTA followed by ultrapure water and equilibrated with the low-salt buffer). Fractions (0.5 mL) were analysed for protein via $A_{280nm}$ and metal by ICP-MS using corresponding matrix-matched calibration curves. To compete Mn$^{II}$ versus Fe$^{II}$ and Cu$^{I}$, stocks were prepared as described earlier and refolding done in an anaerobic chamber. The concentrations of MncA ($A_{280nm}$) and metals (ICP-MS) were superimposed to identify metals co-eluting with MncA and estimate the metal:protein stoichiometry. The ratio of trapped metals was used to determine the relative binding preferences of the two metals at folding in buffer of known competing metal availabilities (Table 1, Supplementary Tables 1, 2).

## (Ni$^{II}$)$_2$MncA crystal structure

A 100 µM solution of MncA in 50 mM HEPES pH 7.5 and 8 M urea, prepared from pelleted inclusion bodies, was added dropwise, with stirring to 100 mL of 50 mM MOPS, pH 7.5, and 10 µM NiSO$_4$, passed through a 0.22 µm filter and loaded on a 5 mL Q-Sepharose column (Cytiva) equilibrated with 50 mM Tris, pH 7.5, and 50 mM NaCl. The column was washed with the same buffer. Folded, concentrated (Ni$^{II}$)$_2$MncA was eluted with buffer containing 500 mM NaCl. Buffer was exchanged via several cycles of dilution and concentration (Amicon Ultracel, 0.5 mL 10 kDa) to obtain MncA (10 mg/mL) in 10 mM Tris, pH 7.5, 50 mM NaCl. Crystals were grown in 100 mM sodium acetate, pH 4.0 and 8–10% PEG 8000[8]. An aliquot of crystals redissolved in 50 mM Tris, pH 7.5, 50 mM NaCl were analysed by ICP-MS and $A_{280nm}$ measured confirming stoichiometric metalation with Ni$^{II}$. Following cryo-protection by the stepwise addition of glycerol to 20% v/v, crystals were flash-cooled and stored in liquid nitrogen prior to data collection.

Data were collected at beamline I04, Diamond Light Source (Supplementary Table 3). A highly redundant data set was collected by obtaining 360˚ of data in four separate scans along the axis of a long, hexagonal rod-shaped crystal. The data were processed using the xia2 package at Diamond[58], which employed XDS to integrate[59], and AIMLESS to scale and merge the data[60]. The structure solution was obtained by molecular replacement with the trimeric (Mn$^{II}$)$_2$MncA structure (PDB ID 2VQA) as search model using Phaser[61] implemented in Phenix[62]. Refinement continued in Phenix alternating with modelling in Coot[63]. Figures were prepared with ChimeraX[64], and PyMOL (Schrödinger) software. MOLE 2.5 was used to identify channels present in the (Ni$^{II}$)$_2$MncA structure[65].

## Expression and purification of RncR

Purification of RcnR overexpressed in *E. coli* BL21(DE3) from coding sequences cloned in pET29a has been described[39,66]. In common with the other regulators (MntR, Fur, NikR, ZntR, Zur, CueR), the sequence was from *Salmonella enterica* serovar Typhimurium strain SL1344 (referred to as *Salmonella*), and RcnR shares 100% sequence identity to *E. coli* RcnR. Anaerobic, reduced and apo-RcnR was prepared by applying purified, EDTA treated, apo-protein to a 1-mL HiTrap heparin column, transferred to an anaerobic chamber, washed with >10 column volumes of Chelex-treated, N$_2$-purged 240 mM KCl, 60 mM NaCl, 10 mM HEPES, pH 7.0, then eluted with 800 mM KCl, 200 mM NaCl, 10 mM HEPES, pH 7.0. RcnR was quantified by $A_{280nm}$ using experimentally determined extinction coefficient of 2,422 M$^{-1}$ cm$^{-1}$ obtained

via quantitative amino acid analysis. Reduced thiol and metal content were assayed[66,67], and all anaerobic protein samples (maintained in an anaerobic chamber) were ≥90% reduced and ≥95% metal-free. All in vitro experiments were carried out under anaerobic conditions using Chelex-treated and $N_2$-purged buffers[66,67].

## Ni[II] stoichiometry and affinity of RcnR

All experiments were conducted in 100 mM NaCl, 400 mM KCl, 10 mM HEPES pH 7.5. To determine stoichiometry, Ni[II] (as $NiCl_2$) was titrated into purified protein (17.2 μM) and absorption spectra recorded at equilibrium using a $\lambda_{35}$ UV-visible spectrophotometer (Perkin Elmer Life Sciences). Additionally, an aliquot of RcnR (20 μM monomer) was incubated with Ni[II] (30 μM) and bound metal resolved by SEC eluted with 100 mM NaCl, 400 mM KCl, 10 mM HEPES pH 7.5 (PD-10, collecting 0.5 mL fractions) and analysed for protein by Bradford assay standardised with known concentrations of RcnR and metal by ICP-MS. To determine Ni[II] affinity, titrations were performed in the presence of EGTA using four RcnR monomer concentrations; 40.4 μM RcnR and 464 μM EGTA, 31.5 μM RcnR and 471 μM EGTA, 25.3 μM RcnR and 479 μM EGTA, 15.3 μM RcnR and 243 μM EGTA, monitoring a Ni[II]-dependent feature of Ni[II]-RcnR at 326 nm. A simultaneous fit was made to all data sets using Dynafit (fitting models in Supplementary Software)[68].

## Ni[II]-RcnR DNA-affinity by fluorescence anisotropy

Fluorescently-labelled (hexachlorofluorescein) double-stranded DNA probes containing the identified RcnR-binding site upstream of the *rcnA* promoter were synthesised and annealed as described[39,66]. Ni[II]-RcnR (1:1 Ni[II]:RcnR tetramer) or apo-RcnR were titrated into 10 nM DNA in 60 mM NaCl, 240 mM KCl, 10 mM HEPES pH 7.5. Changes in anisotropy ($\Delta r_{obs}$) were measured using a modified Cary Eclipse fluorescence spectrophotometer (Agilent Technologies) fitted with polarising filters ($\lambda_{ex}$ = 530 nm, $\lambda_{em}$ = 570 nm, averaging time = 15 s, replicates = 3, and T = 25 °C), allowing the cuvette to equilibrate (3 min) before recording. A simultaneous fit was made to all data sets (fitting models in Supplementary Software using maximum $\Delta r_{obs}$[39,66]).

## Expression and purification of soluble MncA to determine in vivo metalation

The coding region of *mncA* as in pET29a-*mncA* was sub-cloned to create pBAD30-*mncA* to enable tuned expression dependent on [arabinose]. *E. coli* BW25113 (hereafter *E. coli*) containing pBAD30-*mncA* was inoculated into overnight cultures (10 mL LB + 0.2% w/v glucose + carbenicillin at 37 °C, 180 rpm) used the following day to inoculate 2 L flask containing 1 L LB medium and carbenicillin (no glucose) and incubated at 37 °C, 180 rpm, until OD reached mid-log phase ~$OD_{600nm}$ 0.6–0.8, ~3 h. Cultures were transferred to 18 °C and a low concentration (0.02% w/v) L-arabinose added to induce low-level gene expression followed by ~18 h culturing overnight (3 h + 18 h = ~21 h total) before harvesting cells by centrifugation (4000 × g, 4 °C). Purification of soluble in vivo metalated MncA involved protocols analogous to procedures used to recover native MncA from *Synechocystis*[8]. The entire cell pellet (from 1 L culture) was resuspended in 30 mL lysis buffer (20 mM Tris pH 7.5, 1 mM EDTA, 1 mM PMSF) sonicated (4 min pulsing) and centrifuged (27,000 × g, 4 °C) 45 min to remove cell debris. Supernatant was loaded onto a Q-Sepharose anion exchange column (5 mL, pre-equilibrated with 20 mM Tris pH 7.5) then washed with the same buffer. MncA was eluted using a 0–300 mM NaCl gradient in 30 mL, collecting 1 mL fractions. SDS-PAGE identified MncA-containing fractions which were pooled (~3–5 mL). Further purification and analysis used a more rapid analytical protocol in later experiments while in earlier experiments MncA was loaded onto Superdex 75 or Superdex 200 SEC columns (as specified) with fractions analysed by SDS-PAGE. MncA containing fractions were diluted to [NaCl] ≤ 50 mM, reapplied to Q-Sepharose (1 mL) washed then

eluted with 300 mM NaCl and fractions analysed by SDS-PAGE, $A_{280nm}$, and [metal] by ICP-MS. Proportional (%) occupancies of MncA with each metal was first calculated from the ratio of [metal]/[MncA] assuming 2 metal sites per MncA molecule, confirmed by mean experimental occupancy of 99% (replicates 1–3 in Supplementary Table 4a, b). The protocol was simplified to two steps with MncA first recovered via anion exchange (5 mL Q-Sepharose column, 0–300 mM NaCl gradient) followed by rapid analytical scale SEC using a SW3000 (TSK) column[8]. Fractions were again analysed by SDS-PAGE, $A_{280nm}$ and ICP-MS. A comparative fourth biological replicate of MncA extracted from cells grown without metal supplementation was purified via the simplified approach obtaining similar metal occupancies (Supplementary Table 4). Additionally similar occupancies were calculated as a proportion of the total metal content of MncA avoiding variation in MncA $\varepsilon_{280nm}$ with different metals (Supplementary Table 4b). In subsequent metal-supplemented cultures, respective metals were added at inoculation 1–3 h prior to L-arabinose addition. MncA isolated from cells supplemented with copper co-purified with a copper-protein tentatively identified as GAPDH. Fractions containing MncA were passed over a 5 mL Cibacron blue Sepharose (Blue Sepharose) column equilibrated with 50 mM Tris pH 7.5. MncA eluted rapidly while the contaminating protein was retained then eluted with 1 M NaCl in 50 mM Tris pH 7.5. Blue Sepharose-treated MncA-containing solution was subjected to analytical SEC as above.

## Estimation of transcript abundance in *E. coli*

Two extracts (1 mL) were collected from *E. coli*, including from cells containing pBAD30-*mncA* expressing MncA immediately before arabinose addition, and secondly after overnight growth, and RNA stabilised using RNAProtect Bacteria Reagent, 2 mL (Qiagen). Samples were processed as described[27]. Briefly, RNA was extracted using RNeasy Mini Kit (Qiagen), [RNA] estimated from $A_{260nm}$ then treated with DNase I (Fermentas). ImProm-II Reverse Transcriptase System (Promega) generated cDNA, with parallel control reactions excluding reverse transcriptase. Transcript abundance was determined using primers for *mntS*, *fepD*, *rcnA*, *nikA*, *znuA*, *zntA*, *copA* and *rpoD* that amplify ~100 bp of DNA with sequences listed in Supplementary Data 9. Quantitative polymerase chain reaction (qPCR) analysis was executed in 20 μL reactions containing 5 ng of cDNA, 400 nM of each complementary primer and PowerUP SYBR Green Master Mix (Thermo Fisher Scientific). Three technical replicates of each biological replicate were analysed using a Rotor-Gene Q 2plex (Qiagen; Rotor-Gene-Q Pure Detection Software) with additional control reactions without cDNA templates (qPCR grade water used instead, supplied by Thermo Fisher Scientific) run for each primer pair, in addition to control reactions without reverse transcriptase for the reference gene primer pair (*rpoD*). $C_q$ values were calculated with LinRegPCR (version 2021.1) after correcting for amplicon efficiency (Supplementary Source Data TXT, shown in Fig. 7). Change in gene abundance, relative to the control condition (defined as the condition where the minimum transcript abundance was observed for each target gene), was calculated using the $2^{-\Delta\Delta CT}$ method[69] using *rpoD* as the reference gene and presented as $\log_2$(fold change).

## Intracellular metal availabilities and predictions of in vivo metalation

Responses of metal sensors ($\theta_D$ for DNA occupancies of metal-dependent de-repressors and co-repressors, $\theta_{DM}$ for metalated activators) as a function of intracellular available buffered metal concentrations were calculated using sensor metal affinities, DNA affinities, protein abundances and numbers of DNA-binding sites for *E. coli* sensors as described[23] (Supplementary Data 2 for Ni[II]RcnR). Transcript abundance was correlated with the response curves to enable estimations of intracellular metal availability expressed as a free energy for complex formation ($\Delta G_M$) in *E. coli* grown aerobically in LB

as described[27]. Using these availabilities, metalation of proteins was predicted in vivo, accounting for multiple inter-metal competitions including competition from the intracellular buffer as described by Young and coworkers[22]. Supplementary Data 3 and 4 perform these calculations for ideal cells (sensors at mid-range, including $Ni^{II}RcnR$) and at intracellular metal availabilities in *E. coli* grown aerobically in LB respectively.

In metal-supplemented cultures, $log_2$(fold change) relative transcript abundance in this work approximated values reported previously with some boundaries exceeded[27] (Fig. 7). The reported values for high intracellular availabilities of supplemented metals were thus substituted into Supplementary Data 4, while retaining original values for all other metals, to make first predictions of MncA metalation at high intracellular $Mn^{II}$, $Co^{II}$, $Ni^{II}$, $Cu^{I}$ or $Zn^{II}$. For $Mn^{II}$, $Co^{II}$ and $Ni^{II}$, residual differences between predicted and observed MncA metalation was used to iteratively refine intracellular metal availabilities according to Supplementary Note 2 and using Supplementary Data 5. Supplementary Data 6–8 contain the refinements and can be used to predict metalation of other proteins in *E. coli* in elevated $Mn^{II}$, $Ni^{II}$ or $Co^{II}$.

### Elemental analyses by ICP-MS and metal atoms cell⁻¹

ICP-MS was performed at the Durham University Bio-ICP-MS Facility (ThermoFisher iCAP RQ model) with matrix matched standard curves and internal silver standards. *E. coli Δfur* and *ΔmntR* strains were obtained from the Keio collection[70]. Individual colonies of *E. coli* or mutants were inoculated in LB (5 mL) shaking at 37 °C for 3–4 h, diluted into fresh medium (10 mL) in a 50 mL conical centrifuge tube supplemented with metals ($MnCl_2$, $CoCl_2$, $NiSO_4$) where specified, to $OD_{600nm}$ 0.008. Cells were incubated with shaking at 180 rpm, 37 °C, overnight, 100 μL diluted 1:10 to measure $OD_{600}$, and cells recovered (from remaining 9.9 mL) by centrifugation. Pellets were washed four times by resuspension in 1 mL wash buffer (20 mM Tris pH 8.5, 0.5 M sorbitol, 0.2 mM EDTA) followed by centrifugation. Ultrapure $HNO_3$ (Merck) 65% v/v (0.4–0.5 mL) was added to each pellet and allowed to incubate for a minimum of 16 h until fully digested. The samples were then prepared for ICP-MS with matrix-matched calibration curves. An $OD_{600nm}$ of 1 equated to a cell count (CASY cell counter) of $6.47(\pm 0.09) \times 10^8$ cells mL⁻¹ using this strain, enabling atoms cell⁻¹ to be calculated from calibrated ICP-MS data.

### Statistics and reproducibility

Sample sizes were chosen based on prior experimental experience, and to give consistent results, following convention in the literature for equivalent analyses. Experiments designed to derive quantitative values used to predict or test and measure metalation, or to refine and evaluate estimates of intracellular metal availabilities, were performed in triplicate ($n = 3$) or more ($n = 4$) to enable calculation of SD (listed in tables or text or shown as error bars on figures) or SE for $Ni^{II}$-RcnR affinities. Predictions of metalation do not propagate SD from contributing values. Analogous chromatograms to the representative data in Figs. 4b and 5a were obtained on two further occasions ($n = 3$), data in Figs. 4a and 1b are representative of 18 and > 3 analogous purifications respectively. The number of independent experiments or biologically independent samples is otherwise shown in figure legends or footnotes of Tables.

### Reporting summary

Further information on research design is available in the Nature Portfolio Reporting Summary linked to this article.

### Data availability

All data are available within the article, its Supplementary Information files, plus PDB entry 9GOF and from corresponding authors on request. Source data are provided with this paper as Source Data files.

Excel spreadsheets (with instructions) providing a calculator to formulate competing metal buffers, to calculate DNA occupancy as a function of $Ni^{II}$ availability for metal-dependent de-repressor RcnR and providing a calculator to use in vivo recovered metal occupancies of MncA as a probe to refine estimates of intracellular metal availabilities, are provided as Supplementary Data 1, 2 and 5 respectively. Excel spreadsheets constituting calculators of metalation in $Ni^{II}$-RcnR-refined idealised cells, *E. coli* grown aerobically in LB, *E. coli* grown aerobically in LB supplemented with manganese, nickel, and cobalt, are provided as Supplementary Data 3, 4, 6–8 respectively. Supplementary Data 9 contains oligonucleotide sequences. Published structures used here for MncA and MntR are PDB entries 2VQA and 9C4D respectively. Source data are provided with this paper.

### Code availability

Equation derivations are in Supplementary Notes 1 and 2 of the Supplementary Information. Dynafit scripts are provided in Supplementary Software.

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

## Acknowledgements

The Diamond Light Source is acknowledged for time on beamline I04 under proposal MX32736, Prof. Timothy Blower, Prof. Liz Morris for assistance in crystallographic data collection and processing. This work was supported by Biotechnology and Biological Sciences Research Council awards BB/W015749/1 (N.J.R.), Understanding mis-metalation of native versus heterologously expressed protein, and BB/V006002/1 (N.J.R.), A calculator for metalation inside a cell, along with BB/S009787/1 (N.J.R.) supporting networking in Industrial Biotechnology. The authors acknowledge the contributions of Deenah Morton (née Osman) to whom the work is dedicated.

## Author contributions

T.R.Y., S.E.C., A.G., E.T., and N.J.R. contributed to in vivo MncA metal-binding experiments. T.R.Y., A.G., S.E.C., and E.T. contributed to in vitro refolding experiments, with T.R.Y. doing foundation work with S.E.C. and A.G. completing the bulk of these experiments. T.R.Y. derived calculations to formulate competing metal buffers and to use MncA as a probe to refine estimates of intracellular metal availabilities and designed related experimental protocols. A.G. generated the crystal structure of $Ni^{II}$-MncA. P.T.C. and A.J.P.S. analysed $Ni^{II}$ and DNA binding properties of $Ni^{II}$-RcnR. A.G., S.E.C. and E.T. determined metal contents of *E. coli* cells. S.E.C. performed and analysed transcript abundance by qPCR. Relationships between DNA occupancies of metal sensors, intracellular metal availabilities, and qPCR data, used methods established by T.R.Y., N.J.R. T.R.Y. developed the metalation calculations in a form used here. N.J.R., S.E.C., with input from A.G. and E.T., wrote a first draft of the manuscript and generated graphics while all authors edited and approved the final version. T.R.Y., then S.E.C., then A.G., sequentially contributed the bulk of laboratory work at different phases of the programme, with T.R.Y. and S.E.C. making equivalently large contributions. N.J.R. with support of all authors, and especially T.R.Y., S.E.C., and A.G., interpreted the significance of the data. N.J.R. had overall responsibility for the design, finance, and management of the project throughout.

## Competing interests

The authors declare no competing interests.
