## [Transparent Peer Review file · Nature Communications]

A metal-trap tests and refines blueprints to engineer cellular protein metalation with different elements

Corresponding Author: Professor Nigel Robinson

Version 0:

Reviewer comments:

Reviewer #1

(Remarks to the Author)

The manuscript entitled "A metal-trap tests and refines blueprints to engineer cellular protein metalation with different elements" describes the use of MncA to kinetically trap metals within cells to determine metal availability. MncA metal preferences are combined with a model using the responses of metal sensors described in a previous publication to correctly predict metalation (or mismetalation) of MncA under different conditions. The authors also include a characterization of the Ni(II)-responsive RcnR protein to refine the model with respect to Ni(II) availability as well as a high-quality crystal structure of MncA bound to Ni(II). Interesting changes in availability of one metal upon supplementation with another were noted and rationally explained.

Overall, this is a large body of work that uses an innovative approach to further refine metalation predictions. The experiments are carefully performed, interpreted, and validated. The result is a useful tool for predicting metalation of proteins in vivo.

I have no issues with the scientific content of this manuscript. However, it is a somewhat difficult read for the non-expert. I think that a more detailed description of the model in the introduction might be useful. This could include a brief description of how metal sensor transcription levels are used to estimate metal availability and how this is combined with target metal preference to predict metalation. Perhaps even a cartoon summarizing the process would be nice. This might help put the results into context, explaining why relative metal preferences of MncA must first be determined and the reason for RcnR characterization. Otherwise, I have no major revisions to recommend.

Reviewer #2

(Remarks to the Author)

In this manuscript, Robinson and coworkers have designed an experimental system to systematically examine and predict how a protein's intrinsic metal affinity and a metal's intracellular bioavailability together impact metal speciation in biological systems. This research is significant at the fundamental level as it provides novel insight into factors that influence protein metallation in cells, which is of broad interest to bioinorganic chemists that study metals in biology. Their findings also have practical applications to bioengineering/bioprocessing efforts that rely on metalloproteins by providing blueprints for optimizing specific metallation of proteins in biological systems. Their test case is a metal-binding cupin (MncA) from a cyanobacterium that can kinetically trap and coordinate a variety of metals, including Fe(II), Mn(II), Co(II), Cu(I/II), Zn(II), and Ni(II). They first establish the metal binding preferences of the purified apo-protein by folding the denatured protein in the presence of different combinations of metals in metal-buffered solutions. These experiments confirm that the metal binding preferences of MncA follow the Irving-Williams series, exhibiting the lowest affinity for Mn(II) and the highest for Cu(II). Furthermore, analysis of the structure and metal exchangeability of Ni(II)-substituted MncA confirms that non-cognate metals are kinetically trapped in the protein. In previous work that was further refined here, they used data from a series of metal-responsive transcription factors for which they have carefully measured intracellular abundance, transcriptional activity, and metal and DNA affinities, to establish the bioavailability of essential metals in the *E. coli* cytosol. This information correctly predicts mismetallation of MncA with Fe(II) when expressed in *E. coli* in unsupplemented media. However, they clearly demonstrate that in vivo metal loading of MncA can be predictably manipulated by addition of specific metals to the growth media at concentrations that maximize intracellular availability of the metal without causing toxicity. Finally, all this information is incorporated into several calculator spreadsheets that can be used by other researchers to measure metal preferences of a protein of interest and optimize its metal occupancy when expressed in *E. coli*. Overall, the methods are

rigorous, thorough, and detailed, the results clearly represented and convincing, and the analysis and conclusions carefully explained. The writing and manuscript organization are clear and succinct. Although the substantial amount of supplementary material provided makes the study quite dense, the bioinorganic community will surely appreciate the total transparency of the methods, raw data, and calculations contained therein, which can be used as a guide for other researchers. I only have a minor comment to address:

1) In Table 1, note f: the reference to Figure 1b needs to be changed to 1c, and the reference to Supplemental Figure 1a needs to be changed to 1b.

The manuscript has also been edited to address the comments of the reviewers as set-out below.

Reviewer #1 (Remarks to the Author):

I think that a more detailed description of the model in the introduction might be useful. This could include a brief description of how metal sensor transcription levels are used to estimate metal availability and how this is combined with target metal preference to predict metalation. Perhaps even a cartoon summarizing the process would be nice. This might help put the results into context, explaining why relative metal preferences of MncA must first be determined and the reason for RcnR characterization.

This is a helpful suggestion since we are especially keen for the work to be accessible to a broad audience. The first sentence of the Introduction (line 27), and again in line 74 relating to how the metal sensors are used, now refer to a new Supplementary Figure 1. The new Figure and its legend include the suggested diagrams and explanations. Previous Supplementary Figures 1 and 2 have been combined into a revised Supplementary Figure 2. The new legend also cites two mini reviews.

Reviewer #2 (Remarks to the Author):

1) In Table 1, note f: the reference to Figure 1b needs to be changed to 1c, and the reference to Supplemental Figure 1a needs to be changed to 1b.

This has been corrected noting that the new Supplementary Fig. 1, changes 1b to 2b.